

# Process-Based Flood Frequency Analysis in an Agricultural Watershed Exhibiting Nonstationary Flood Seasonality

Guo Yu[1], Daniel B. Wright[1], Zhihua Zhu[2], Cassia Smith[3], Kathleen D. Holman[4]

[1]Department of Civil and Environmental Engineering, University of Wisconsin-Madison, Madison, 53706, USA
[2]Department of Water Resources and Environment, Sun Yat-sen University, Guangzhou, 510275, China
[3]Department of Civil and Environmental Engineering, Carnegie Mellon University, Pittsburgh, 15213, USA
[4]Research and Development Office, Bureau of Reclamation, Denver, 80225, USA

*Correspondence to*: Zhihua Zhu (zzhu264@wisc.edu)

**Abstract.** Floods are the product of complex interactions of processes including rainfall, soil moisture, and watershed morphology. Conventional flood frequency analysis (FFA) methods such as design storms and discharge-based statistical methods offer few insights into process interactions and how they "shape" the probability distributions of floods. Understanding and projecting flood frequency in conditions of nonstationary hydroclimate and land use requires deeper understanding of these processes, some or all of which may be changing in ways that will be undersampled in observational records. This study presents an alternative "process-based" FFA approach that uses stochastic storm transposition to generate large numbers of realistic rainstorm "scenarios" based on relatively short rainfall remote sensing records. Long-term continuous hydrologic model simulations are used to derive seasonally varying distributions of watershed antecedent conditions. We couple rainstorm scenarios with seasonally appropriate antecedent conditions to simulate flood frequency. The methodology is applied in Turkey River in the Midwestern United States, a watershed that is undergoing significant climatic and hydrologic change. We show that using only 15 years of rainfall records, our methodology can produce more accurate estimates of "present-day" flood frequency than is possible using longer discharge or rainfall records. We found that shifts in the seasonality of soil moisture conditions and extreme rainfall in Turkey River exert important controls on flood frequency. We also demonstrate that process-based techniques may be prone to errors due to inadequate representation of specific seasonal processes within hydrologic models. Such mistakes are avoidable, however, and our approach may provide a clearer pathway toward understanding current and future flood frequency in nonstationary conditions compared with more conventional methods.

## 1 Introduction

Riverine floods, among the most common natural disasters worldwide, are the product of complex interactions between heavy rainfall, watershed and river channel morphology, and antecedent conditions including soil moisture and snowpack. Their impacts are projected to increase in the future due to hydrometeorological factors (e.g. Hyndman, 2014) and increased human development in flood prone areas (e.g. Ntelekos et al., 2010; Ceola et al., 2014; Prosdocimi et al., 2015). Estimating





relationships between flood likelihood and severity is central to flood risk management and infrastructure design; these relationships are typically represented by flood frequency distributions (or curves), while the broad family of procedures used to derive them is termed flood frequency analysis (FFA). Most existing FFA methods belong to one of three approaches: statistical analysis of observed streamflow, design storms, or continuous simulation and other so-called "derived" or "process-based" methods. Each has strengths and shortcomings, which are briefly summarized in Sect. 2 (see Wright et al., 2014a for a distinct summary).

FFA is challenging even in stationary (i.e. unchanging) watershed and hydroclimatic conditions due to the scarcity of observations of large floods and the associated factors that generate them (Stedinger and Griffis, 2011). The role of soil moisture in flood frequency, for example, is very important (Berghuijs et al., 2016), but poorly understood due to a lack of long-term observations. Furthermore, the individual and joint flood causative factors will evolve as a watershed undergoes changes in land use or hydroclimate (Machado et al., 2015). Leading causes of change (i.e. nonstationarity) include human intervention (Konrad and Booth, 2002; Schilling and Libra, 2003; Villarini et al., 2009), natural climate variability (Enfield et al., 2001; Jain and Lall, 2000) and climate change driven by increased greenhouse gases (Milly et al., 2008; Hirsch and Ryberg, 2012). Combinations of these will lead to nonstationary flood frequency, a challenge for which the bulk of existing FFA methods are ill-suited (El Adlouni et al., 2007; Gilroy and McCuen, 2012).

In this study, we present an alternative FFA methodology that aims to "construct" the flood frequency curve through a combination of observations, stochastic methods, and hydrological modeling that generates and combines the causative factors (i.e. processes) such as rainfall and soil moisture that produce floods. This concept is not new, and has traditionally be called "derived FFA" (e.g. Eagleson, 1972; Franchini et al., 2005; Haberlandt, 2008), though we prefer the more descriptive term "process-based FFA" (after Sivapalan and Samuel, 2009; see Clark et al., 2015a, 2015b and Lamb et al., 2016; who discuss somewhat similar techniques).

We apply our process-based methodology to an agricultural watershed in the Midwestern United States that is undergoing substantial seasonal hydroclimatic and hydrologic changes that have led to nonstationary flood frequency. We will show that process-based FFA may hold better prospects than other methods in this watershed and more broadly, and is useful for deciphering the underlying physical drivers of flood frequency. (The reader is directed to Sivapalan and Samuel (2009) for a strong argument in favor of process-based approaches in the face of nonstationary conditions, though they do not actually lay out a specific FFA procedure.) Our methodology underscores the importance of seasonality in the joint contributions of rainfall and soil moisture to flood frequency. To our knowledge, this study is the first to explore the role that seasonal changes in hydroclimatic and hydrologic processes play in nonstationary flood frequency. We also argue that any process-based FFA approach will require careful consideration of seasonality.

The structure of the paper is as follows: Section 2 briefly reviews three broad types of FFA approaches. Section 3 introduces the study region, watershed, and hydrometeorological data. Section 4 outlines the process-based FFA methodology used in this study, including the hydrological model, stochastic storm transposition (SST) procedure used to derive the synthetic rainfall scenarios, and elements of both continuous and event-based rainfall-runoff simulation. The nonstationary hydroclimate





of the study watershed and trends in relevant hydrometeorological variables are analyzed in Sect. 5.1. Process-based FFA results are presented and compared with "conventional" statistical estimates in Sect. 5.2. Simulated flood seasonality is explored in Sect. 5.3. The relationship between rainfall and simulated peak discharge quantiles is examined in Sect. 5.4. Section 6 includes a summary and concluding remarks.

## 2 Review of FFA Approaches

### 2.1 Discharge-based Statistical Approaches

Statistical FFA approaches involve fitting a statistical distribution to extreme discharge observations and extrapolating this distribution to estimate quantiles such as the 100-year or 500-year discharge. While these approaches utilize direct observations of flooding (e.g. peak discharge or volume), long streamflow records at or near the given river cross section needed for reliable quantile estimates. Such records are lacking in many locations, even in developed countries. Statistical approaches are limited by the available observations; thus, the estimation distribution not represent the "true" (unknown) distribution of possible outcomes (Linsley, 1986; Klemeš, 1986, 2000a, 2000b). Regional FFA methods are able to improve quantile estimates both at gaged and ungauged locations (Dawdy et al., 2012); however, they make assumptions regarding the transferability of regional information to specific locations and can, in doing so, may neglect key geophysical processes that dominate the spatiotemporal variability of floods (Ayalew and Krajewski, 2017).

Though streamflow observations are the result of a range of complex factors including rainfall, soil moisture, and channel routing, without concurrent observations of these "upstream" variables, neither streamflow observations nor distributions fitted to them provide much insight into flood causes. Long-term records of such variables, particularly soil moisture, are virtually nonexistent. There have been numerous examples within the FFA literature pointing to situations in which discharge-based analyses are inferior to others based on hydrologic modeling, including cases of basin storage "discontinuities" (Rogger et al., 2012), reservoirs (Ayalew et al., 2013), and land use change (Cunha et al., 2011).

Finally, most statistical FFA methods assume that the magnitude of extreme flood events and quantiles are stationary. This assumption conflicts with numerous examples in which hydrological records exhibit various types of nonstationarity (e.g. Salas and Obeysekera, 2014; Potter, 1976; Villarini et al., 2009; Douglas et al., 2000; Franks and Kuczera, 2002). Though nonstationary statistical FFA techniques do exist (e.g. Cheng et al., 2014; Gilleland and Katz, 2016; Serago and Vogel, 2018), they face severe limitations extrapolating to future conditions (Luke et al., 2017; Sivapalan and Samuel, 2009; Stedinger and Griffis, 2011) since they rarely consider the fundamental physical causes of change.

### 2.2 Design Storm Approaches

Design storm (DS) approaches use idealized rainfall scenarios of a given return period as inputs to a calibrated hydrological model to simulate flood peaks. DS is widely used in practice due to its simplicity (Cudworth, 1989; Kjeldsen, 2007; Ball et al., 2016). To some extent, the flood-producing physical processes are captured via the hydrological model, which also



provides a flood hydrograph as opposed to only the peak discharge or volume provided by statistical approaches. However, DS approaches rely on at least three major assumptions: (1) point-based rainfall intensity-duration-frequency (IDF) estimates (which are subject to some of the same aforementioned statistical and data availability issues as flood discharges) can be converted into hyetographs using dimensionless temporal rainfall distributions and into basin-averaged estimates using area

reduction factors (e.g. Svensson and Jones, 2010); (2) IDF estimates, based on annual rainfall maxima, produce flood peaks which are quantiles of the flood annual maxima distribution; and (3) there is a 1:1 equivalence between rainfall and simulated discharge quantiles (i.e. return periods), for example, a 100-year idealized rainfall event will produce a reasonable estimate of the 100-year peak discharge. The last of these discounts the possibility that watershed initial conditions such as soil moisture and snowpack can modulate the transformation of rainfall quantiles into discharge quantiles.

These assumptions are not without their shortcomings. Wright et al. (2014b), for example, showed significant disparities between observed point and basin-averaged rainfall extremes that cannot be captured using conventional ARF concepts. Viglione and Blöschl (2009) and Vigligone et al. (2009), meanwhile, demonstrated that the ratio of rainfall return period to flood return period is controlled by storm duration, a runoff coefficient (which is related to antecedent conditions), and a runoff threshold effect using design storm approach in conjunction with derived distribution approach. These initial conditions can

vary substantially by season, meaning that high soil moisture may only very infrequently coincide with extreme rainfall. Wright et al. (2014a) discusses additional design storm shortcomings including time of concentration concepts, while also pointing out that design storm approaches.

## 2.3 Continuous Simulation and Process-Based FFA Approaches

Continuous simulation (CS) and process-based approaches to FFA leverage the potential benefits of hydrological models while

minimizing the simplifying assumptions of DS methods. These approaches typically use long series of historical or stochastically generated rainfall, temperature, and occasionally other meteorological variables as hydrological model inputs, to simulate long discharge time series. Peak flows can be extracted from these series and the flood frequency distribution can be obtained. Thus, event rainfall return period and duration and antecedent conditions do not need to be specified and the equality between the rainfall and discharge return period is not assumed (Calver et al., 1999, 2009). In addition, projections of

future flood frequency can be developed by incorporating general circulation model (GCM) rainfall and temperature projections into the input meteorological series (Gilroy and McCuen, 2012; Rashid et al., 2017). On the other hand, CS approaches are limited by the general lack of reliable long-term time series of extreme rainfall and other meteorological data (Blazkova and Beven, 1997, 2002, 2009) and, in the case of sophisticated distributed approaches, by potentially high computational demands (Li et al., 2014; Peleg et al., 2017). Stochastic rainfall generation techniques typically struggle with

producing the extremes that are critical for flooding (e.g. Cameron et al., 2000; Furrer and Katz, 2008), and training such models for locations with rainfall nonstationarities and strong seasonal variations is nontrivial. Camici et al., (2011) and Li et al. (2014) present FFA process-based approaches that couple long CS simulation results with event-based simulations.



One argument in favor of CS and process-based approaches is that the complex joint relationships between flood drivers such as rainfall and soil moisture are resolved within the modeling framework and thus do not rely on users' assumptions. We demonstrate that caution is needed in the representation of seasonality; to briefly summarize, it is critical that both seasonality in input variables as well as seasonally varying processes within the model be "correct." Without verifying this, process-based

approaches may produce incorrect results, or, as shown in Sect. 5, seemingly correct results as a result of incorrect methods.

## 3 Study Region and Data

The study watershed of Turkey River (Fig. 1) is situated in northeastern Iowa and the portion upstream of the US Geological Survey (USGS) stream gage at Garber (ID code 05412500) has a drainage area of 4002 km$^2$. Elevation ranges from approximately 426 m above sea level (masl) in the west to 197 masl at the stream gage site. The streams at the upper part of

10 the catchment have relatively low slopes, while the channels and hillslopes in the lower part are steeper. Soils in Turkey River are mainly loams and silts (IFC, 2014). According to USGS 2012 National Land Cover Dataset (NLCD), Turkey River watershed is predominantly agricultural with less than 2% urban land cover. Comparison of NLCD from 1992, 2001, 2006, and 2012 indicates that land uses have not evolved significantly over time (results not shown), though the hydrologic impacts of subsurface tile drainage, which has become ubiquitous throughout the region, are poorly understood (see, e.g. Schilling et

al., 2014).





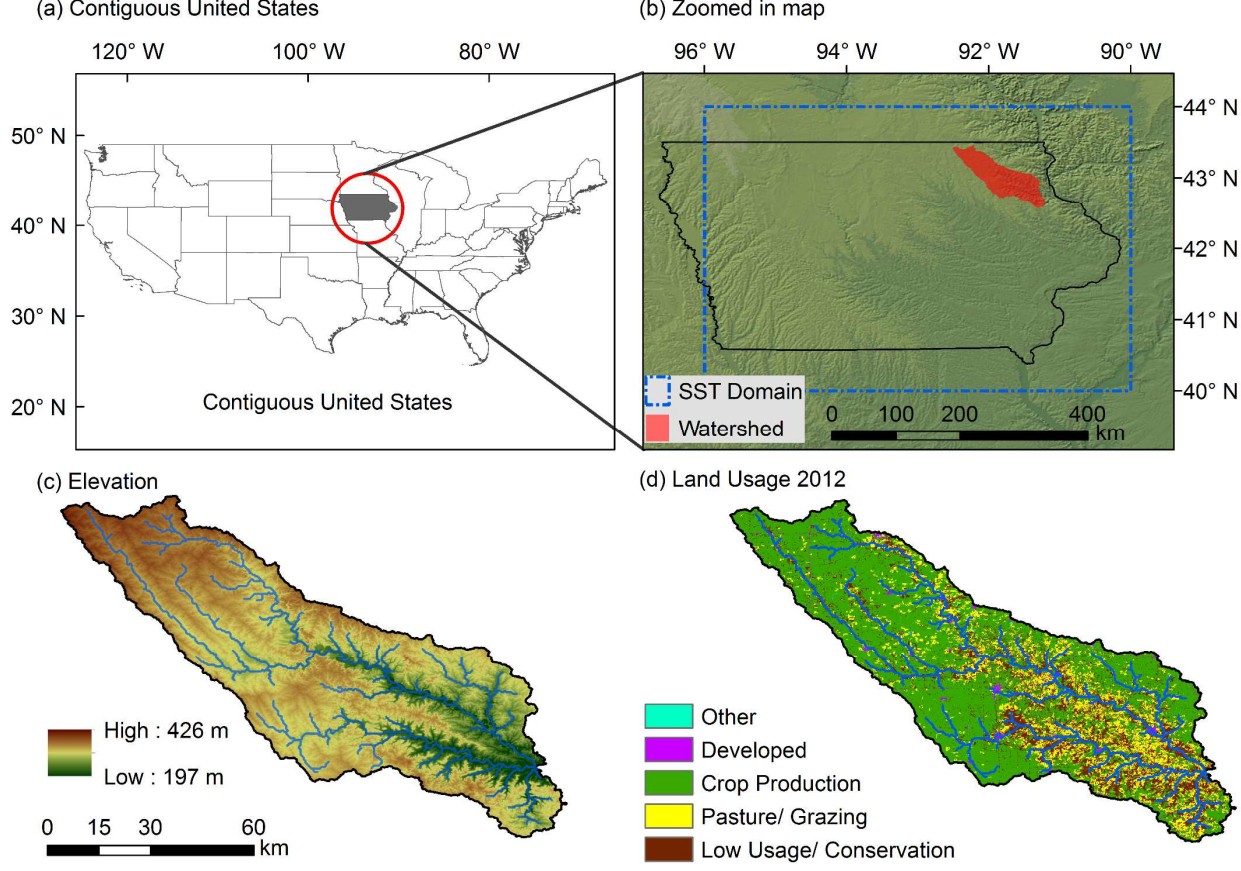

**Figure 1.** Study region. **(a)** Contiguous United States; the state of Iowa is highlighted in grey. **(b)** The zoomed in map shows Turkey River watershed (red) and the extent of the stochastic storm transposition region (blue dash line). **(c, d)** Turkey River watershed showing land surface elevation (based on the USGS National Elevation Dataset) and land use (based on the USGS 2012 NLCD).

5    We use daily discharge observations for 84 years (1933-2016) from the USGS streamgage at Garber, Iowa (USGS gage identifier 05412500) to understand the hydroclimatology of flooding and to validate our FFA results. Daily discharge observations for 69 years (1948-2016), in conjunction with Global Historical Climate Network (GHCN) daily temperature data are used to configure, calibrate, and validate the hydrological model, as described in Sect. 4.1. The CPC US Unified (CPC-Unified; Chen et al., 2008) and Stage IV (Lin and Mitchell, 2005) precipitation data, available through the National

10   Oceanic and Atmospheric Administration, are used for rainfall analyses. CPC-Unified provides daily, 0.25° rainfall estimates interpolated from rain gage observations, while Stage IV provides hourly, approximately 4 km estimates by merging data from rain gages and the National Weather Service Next-Generation Radar network (Crum and Alberty, 1993). In this study, analyses based on Stage IV use data from 2002-2016, while long-term analyses based on CPC-Unified use data from 1948-2016.



## 4 Methodology

The FFA approach presented in this study combines continuous simulation (CS), stochastic storm transposition (SST) using the RainyDay software, and event-based simulation. CS provides large samples of seasonally varying antecedent conditions including soil moisture and snowpack. SST produces large numbers of synthetic rainfall scenarios including realistic estimates
5  of rainfall space-time structure. Together, these drive event-based simulations to generate the synthetic flood peaks that are used to derive flood frequency distributions. The approach is illustrated schematically in Fig. 2 and summarized in the following subsections.





**Continuous Simulation (Section 4.1)**

Long CS rainfall-runoff modelling

Model calibration

Simulated daily streamflow

No — Optimized parameters

Yes

Historical CS simulation

Antecedent conditions: Soil moisture

Grided rainfall data

**Stochastic Storm Transposition (Section 4.2)**

SST software: RainyDay

Generate rainfall scenarios through temporal resampling and spatial transposing of observed storms

Assigning a new event date for each simulated rainfall event (Step 5 of RainyDay in Section 4.2)

20 realizations of 500 years synthetic rainfall event scenarios

**Event-based Simulation (Section 4.3)**

Read one realization of rainfall scenarios

Extract 1 synthetic year rainfall events

Select the 1 year long initial conditions based on assigned date

Run event-based RR model for each rainfall event with its paired initial condition

Obtain the simulated annual peak within a synthetic year

Derive the 500-year FFA

20 realizations

500 synthetic years

20 realizations of 500-year FFA



**Figure 2.** Flow chart showing the process-based FFA approach. Dotted outlines delineate components associated with subsections 4.1, 4.2 and 4.3.

## 4.1 Hydrological Model, Calibration, and Continuous Simulation

We use the lumped Hydrologiska Byråns Vattenavdelning (HBV) model (Bergström, 1992, 1995; Lindström et al., 1997).
HBV has been widely used to study hydrological response in United States (Vis et al., 2015; Niemeyer et al., 2017) and other regions of the world (Harlin and Kung, 1992; Osuch et al., 2015; Seibert, 2003; Chen et al., 2012). The "HBV-Light" ( henceforth referred to as HBV; Seibert and Vis, 2012) version is used in this study, and consists of four main routines: snowpack, soil moisture, catchment response, and runoff routing. The model simulates daily discharges based on time series of precipitation and air temperature, as well as estimates of long-term daily potential evapotranspiration.

The process-based FFA methodology employed in this study could be used with any hydrological model. Utilizing a distributed hydrological model would allow for more realistic representation of important characteristics like changing land use, rainfall spatiotemporal structure, and flood wave attenuation in river channels. Other models could operate higher (i.e. subdaily) resolution in terms of inputs, model time steps, and outputs. We selected HBV at the daily time step due to its simplicity, computational speed, and its ability to represent conceptually multiple watershed hydrological processes.

We calibrated the model automatically via 5000 continuous simulations from 1948-2016 by maximizing Kling and Gupta Efficiency (Gupta et al., 2009) using the Genetic Algorithm and Powell optimization method (Seibert, 2000). After the genetic algorithm has finished, 1000 additional runs are performed for fine-tuning using Powell's quadratic convergent method (Press, 1996). Lastly, the optimized parameter set is manually adjusted to improve the fits between observed and simulated annual peak flow (see Lamb, 1999). More elaborate calibration and uncertainty estimation procedures such as Generalized Likelihood Uncertainty Estimation (GLUE; Beven and Binley, 1992; Beven, 1993; Beven and Binley, 2014) could be used, but are outside the scope of our study. After calibration, HBV was used to perform CS with historical CPC and Stage IV rainfall and temperature data to derive long-term simulated soil moisture and snowpack values, which are usually difficult to obtain via measurement. We "pair" samples of these initial conditions with synthetic rainfall events, as described in Sect. 4.2 and Sect. 4.3.

## 4.2 Stochastic Storm Transposition

Stochastic storm transposition (SST) is a method to generate realistic probabilistic rainfall scenarios through temporal resampling and spatial transposing of observed from the surrounding region. SST is a bootstrap method that aims to effectively "lengthen" the rainfall record by performing "space-for-time substitution" within a rigorous probabilistic framework. Unlike rainfall IDF, SST can preserve observed rainfall space-time structure, and, unlike design storm methods, obviates the need to equate rainfall duration to catchment response time (Wright et al., 2013, 2014a, 2014b). Alexander (1963), Foufoula-Georgiou (1989), and Fontaine and Potter (1989) provide general descriptions of SST. Wilson and Foufoula-Georgiou (1990) apply the





method for regional rainfall frequency analysis while Gupta (1972), Franchini et al. (1996), England et al. (2014) and Nathan et al. (2016) use it for FFA.

Wright et al. (2013) used SST with a 10-year high-resolution radar rainfall dataset to estimate spatial IDF relationships. Wright et al. (2014a) used this approach with a physics-based distributed hydrologic model for FFA in a heavily urbanized watershed,
demonstrating its usefulness in evaluating multi-scale flood response.

RainyDay is an open-source, Python-based SST software that couples SST methods with rainfall remote sensing data. A more detailed description can be found in Wright et al. (2017); not all of its features are used in this study. The following steps describe how RainyDay is modified and used in this study:

1. We define a 6-degree (longitude) by 4-degree (latitude) geographic transposition domain (40° to 44° N, 90° to 96° W; blue dash line of Fig. 1 inset) which encompasses the Turkey River watershed. This same domain was used in Wright et al (2017) and, importantly for the SST approach, extreme rainfall properties are roughly homogeneous within it.

2. The RainyDay software creates a "storm catalog" from 15 years of Stage IV (69 years of CPC) rainfall data that consists of the 450 (2070) most intense rainfall events within the transposition domain. These storms have a maximum duration of 96 hours and must be separated by at least 24 hours. Storms that exhibit "radar artifacts" such as major bright band contamination or beam blockage are excluded from subsequent steps.

3. The RainyDay software generates a Poisson-distributed integer $k$ that represents a "number of storms per year." The rate parameter $\lambda$ of this Poisson distribution is calculated by dividing the total number of rainfall events in the storm catalog by the number of years in the historical rainfall record (i.e. $\lambda = 450/15 = 30.0$ storms per year).

4. RainyDay randomly selects $k$ storms from the storm catalog and transposes the associated rainfall fields within the transposition domain by an east-west distance $\Delta x$ and a north-south distance $\Delta y$, where $\Delta x$ and $\Delta y$ are drown from a 2-dimentional Gaussian kernel density estimate based on the locations of the original storms in the storm catalog. For each of the $k$ transposed storms, the time series of rainfall over the Turkey River watershed is computed. Steps 3 and 4 can be understood as temporal resampling of observed rainfall events to "synthesize" a hypothetical year of rainfall events over the transposition domain and, by extension, over the watershed. Although the rainfall events for the "synthetic" year do not form a continuous series, the dates associated with each storm event are recorded, thus facilitating seasonally consistent flood simulations.

5. All $k$ events within a synthetic year are assigned a new, randomly selected year from 1948-2016 (2002-2016) for CPC (Stage IV) rainfall data, used to select antecedent conditions. This ensures that the $k$ rainfall events are all "embedded" within a realistic annual representation of watershed conditions. Antecedent conditions are randomly selected from +/- 7 days of the updated storm date to ensure realistic seasonality of storms and watershed conditions. A storm that occurred on July 15, 2016, for example, could be paired with initial conditions selected from a day ranging between July 8-22 from a randomly selected year, while the remaining $k$-$1$ events would be paired with seasonally appropriate initial conditions from that same year.





6.  RainyDay repeats Steps 3-5 500 times to create one realization of 500 synthetic years of rainfall events for Turkey River. Twenty realizations are generated. Unlike in the existing version of RainyDay, all rainfall events within a synthetic year are retained for subsequent event-based flood simulations, since the modulating effects of antecedent conditions mean that the largest rainfall event in a given year does not necessarily produce that year's largest flood peak (this possibility is explored in Sect. 5.4).

## 4.3 Event-Based Flood Simulation

Using the "paired" SST-based rainfall events and watershed initial conditions derived from CS (Sect. 4.2), HBV simulates the "event peak" (the maxima daily discharge). The largest "event peak" among the $k$ events within a synthetic year represents the simulated annual maximum daily streamflow. This process is repeated for all 500 synthetic years within each realization, resulting in 500 annual maximum streamflow values, which are then ranked in descending magnitude. The annual exceedance probability $p_e$ (i.e. the probability in a given year that an event of equal or greater intensity will occur) of each maximum streamflow are calculated by dividing its rank by 500 (the total number of simulated annual maximum daily streamflow). The 20 realizations provide estimates of variability for each flood quantile.

## 5. Results

### 5.1 Hydroclimatology and Nonstationarity

Four distinct time periods (Fig. 3a) are used for analyzing the changing hydroclimatology in Turkey River: the USGS daily mean streamflow period of record (1933-2016), a more recent period of apparent elevated flood activity (1990-2016), the period of the Stage IV rainfall record (2002-2016), and the period of the CPC rainfall record (1948-2016). Results here and in subsequent subsections "align" with one or more of these time periods.

The hydroclimate of Turkey River is changing, as shown by the Mann-Kendall (MK) test for monotonic trends (Mann, 1945) and the Thiel-Sen estimator (Sen, 1968), nonparametric methods to determine trend direction and magnitude (i.e. slope), respectively. Since 1948, annual precipitation and discharge show significant increases ($p<0.05$) and their variability has also increased (Table 1), while annual maximum daily discharge has decreased, though not significantly. It is important to note, however, that there are two counteracting seasonal trends (see also Fig. 3a): annual daily discharge maxima in March-April has decreased significantly while May-September has increased somewhat. Thus, the lack of significant change in flood magnitude in Turkey River at the annual scale masks changes in the seasonality of flooding.



**Table 1.** Mann-Kendall trend (two sided) test for hydrological variables. *p*-values are given in parentheses. The variance refers to the absolute residuals associated with the Thiel-Sen estimator.

| Data | Time Range | Trend |
|---|---|---|
| Annual Discharge | 1933-2016 | ↑ (0.001) |
| Annual Max. Daily Discharge | 1933-2016 | ↓ (0.447) |
| Variance of Annual Max. Daily Discharge | 1933-2016 | ↑ (0.056) |
| Annual Max. Daily Discharge in March-April | 1933-2016 | ↓ (0.002) |
| Annual Max. Daily Discharge in May-September | 1933-2016 | ↑ (0.089) |
| Annual Precipitation | 1948-2016 | ↑ (0.003) |
| Annual Max. Daily Precipitation | 1948-2016 | ↑ (0.362) |
| Annual Max. 4-day Precipitation | 1948-2016 | ↑ (0.419) |
| Annual Mean Temperature | 1948-2016 | ↓ (0.462) |
| March-May Mean Temperature | 1948-2016 | ↑ (0.443) |

We examine this flood seasonality, both in observations and in our continuous HBV simulations (Fig. 3b). The seasonal distribution of flood occurrence for 1948-2016 shows a March-April maximum, with elevated flood activity continuing through

May and June. This is distinct from, though overlaps somewhat with both the 4-day annual maxima of rainfall, which occur most frequently in the June-September period, and simulated daily annual maxima soil moisture, which tend to occur in January-April. These results highlight that flood activity is the product of seasonal variations in both soil moisture and rainfall. 4-day rainfall is used in SST; seasonality in 1-day rainfall is very similar (results not shown).

The March-April peak of flood occurrence corresponds with relatively high soil moisture associated with snowmelt, rain on

or frozen soil, or frequent spring rains. The secondary peak of flood occurrence in May-June is associated with larger flood magnitudes (including the largest flood event) due to recent severe thunderstorms. For instance, widespread flooding in Iowa in June 2008 showed that thunderstorm systems make critical contributions to the upper tail of flood peak distributions (Smith et al., 2013). Although the frequent heavy rainfall events in August or September have not triggered any of the recorded annual flood peaks in Turkey River, our process-based FFA demonstrates that they may still relevant to current and future flood

frequency, as shown in Sect. 5.3.

The largest annual maxima (over 800 m³/s) occur in May-July (Fig. 3c), consistent with the broader climatology of flooding in Iowa (Smith et al., 2013; Villarini et al., 2011). Furthermore, both the seasonality and magnitude of flood peaks have shifted since approximately 1990 (Fig. 3a, 3c), with March-April (May-September) floods decreasing (increasing) in magnitude, leading to a shift in the seasonality of the overall distribution of annual maxima daily streamflow from a high in March prior

to 1990 to a prolonged high from April-June post-1990. Although the small sample size of the annual maxima daily discharge during this elevated late-spring/summertime flood period (1990-2016) may affect the reliability of the derived PDF of flood occurrence, Park and Markus (2014) reported a significant shift toward summertime flooding in the nearby Pecatonica River. Statistically based FFA (including nonstationary methods) based on annual maxima discharges may fail to capture the impact of this shifting seasonality on flood frequency.



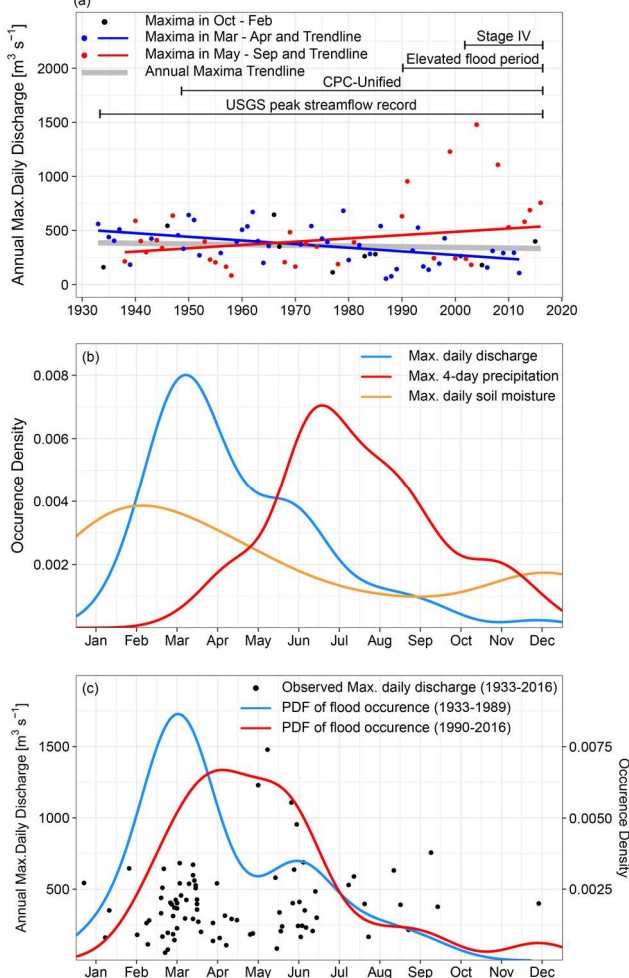

**Figure 3. (a)** Linear trends for two groups of annual maxima daily discharge: March-April floods (blue) and May-September floods (red). The October-February maxima daily discharge are in black dots and its trend line is not calculated because only nine annual maxima are during this period. The trend line for the "overall" annual maxima time series (i.e. disregarding seasonality) is in wide grey line. Four critical time ranges are shown in black lines. **(b)** Occurrence densities of the date during the year for the observed annual daily maxima discharge, observed annual 4-day maxima precipitation, and simulated annual daily maxima soil moisture in Turkey River watershed from 1948 to 2016. **(c)** The magnitude and the date during the year for annual flood peaks in Turkey River at Garber is in black dot. Sample probability density functions for flood events (1933-1989, blue; 1990-2016, red) are shown. In this study, all probability densities for occurrence date are estimated using a Gaussian kernel smoother.

As mentioned in Sect. 2.3, caution is needed when using hydrological models for process-based FFA. The hydrological model must be faithful, to a reasonable degree, to observed nonstationarities, while the importance of soil moisture in flooding implies that processes such as subsurface flow and storage beyond the event scale must also be adequately represented. "Quantile-Kendall plots" (Hirsch and De Cicco, 2015) for observed and simulated daily streamflows from 1948 to 2016 highlight this (Fig. 4). Each point on the plot is a trend slope computed for the 1948-2016 period using the Thiel-Sen estimator for a given quantile of the variable in question, while the color of the point indicates the significance of the trend computed using the MK





test. For instance, the point at the far left (right) is the first (365[th]) order statistic, which is the annual minimum (maximum). The plots are useful for displaying long-term trends across the entire distribution.

Trends in observed streamflow (Fig. 4b) below the 90[th] percentile are largely positive (around 1.5% per year) and significant at the 5% level. Beyond the 90[th] percentile, the trend is less significant. Quantile-Kendall plots for simulated daily streamflow derived and without the HBV snowpack routine (Fig. 4c, 4d) reveal that simulated streamflow trends without the snowpack routine more closely resemble the observed trends, since the trend slope generally decreases with increasing quantile and the significance at high quantiles is generally low. (Note that model calibration is performed separately for both simulations, meaning the model parameters differ between them.) The plot of simulated streamflow with the snowpack routine differs substantially from observations, including a significant 1.5% per year increase in simulated annual maxima which contrasts with an insignificant 0.2% observed decrease. While a different hydrological model structure would produce different outcomes, these results highlight that certain process representations in models may produce undesirable results that could propagate through to FFA.

We also show Quantile-Kendall plots for observed daily precipitation (Fig. 4a) and simulated daily soil moisture derived without (with) the snowpack routine (Fig. 4e, 4f). Both Quantile-Kendall plots for observed precipitation and simulated soil moisture exhibit positive trend slopes that decrease moderately with increasing quantile. It can be inferred from these plots that the increases in precipitation and soil moisture appear to result in an increase in low and moderate flows across the Turkey River watershed, though their implications for flood seasonality are less clear.

Previous studies also have shown increases in annual and seasonal precipitation and streamflow totals as well as changes in the frequency of intense rain events and the seasonality of timing of precipitation in the Midwestern United States and have suggested potential causes including large-scale climate variability and climate warming (e.g. Gupta et al., 2015; Mallakpour and Villarini, 2016, 2015; Park and Markus, 2014; Yang et al., 2013). Specific attribution of the changes in Turkey River is beyond the scope of this study, but these trends nonetheless highlight the potential challenge and important considerations for FFA in a changing hydroclimate.



**Figure 4.** Quantile-Kendall plots for observed precipitation **(a)**, observed daily streamflow **(b)**, simulated daily streamflow with (without) snowpack routine **(c, d)**, and simulated daily soil moisture with (without) snowpack routine **(e, f)** for 1948-2016. The color represents the p-value for the Mann-Kendall test. Red indicates a trend that is significant at 0.05 level. Black indicates an attained significance between 0.05 and 0.1. Grey dots indicate trends that are not significant at the 0.1 level.

### 5.2 Flood Frequency Analyses

RainyDay-based FFA for Turkey River at Garber using both Stage IV and CPC rainfall datasets are compared with statistical discharge-based FFA using 1933-2016 USGS annual maxima daily streamflows (Fig. 5). The latter is derived using the HEC-SSP software (Bartles et al., 2016), which implements methods from USGS Bulletin 17B (Interagency Advisory Committee on Water Data, 1982) using "station skew" to fit the log-Pearson Type III distribution. Observed annual maxima daily streamflow from 1933 to 2016 are also shown, where $p_e$ is estimated using the Cunnane plotting position (Cunnane, 1978). Different HBV parameters are used for the Stage IV and CPC-based simulations, due to different time range and error properties in these datasets. We did not use the snowpack routine in HBV to generate the results in Fig. 5; this routine was shown to produce unrealistic streamflow results (Fig. 4c, 4d).

The Stage IV-based flood frequency curve agrees reasonably well with the discharge-based FFA for $p_e > 0.3$ (left panel of Fig. 5), but higher estimates for $p_e < 0.3$. The CPC-based curve, on the other hand, matches closely with the discharge-based curve. The Stage IV analyses use shorter but more recent (2002-2016) meteorological and hydrological records than the other frequency curves. When the streamflow observations are divided into two groups (1933-1989 and 1990-2016), it becomes

clear that the recent peak flood observations align with the Stage IV-based SST results (right panel of Fig. 5). This, along with the increasing trend of annual mean precipitation and discharge shown in the previous subsection, suggests that the Stage IV-based FFA adequately reflects flood frequency in the wetter recent climate (a similar result is shown in Wright et al., 2017), while the longer CPC-based and discharge-based methods fail to do so.

Considering Fig. 3 and Fig. 4, the results shown in Fig. 5 suggest that the recent shift from spring to summer flood activity is

accompanied by a substantial shift in the flood frequency distribution. The close agreement between CPC-based and discharge-based FFA suggests that even in stationary situations with long records, the statistical methods do not necessarily produce superior results to process-based approaches. We also derived the RainyDay based FFA using CPC-Unified rainfall data from 2002 to 2016 and it closely resembles the Stage IV-based FFA (results not shown), pointing to rainfall temporal nonstationarity, rather than differences between different rainfall datasets, as the primary driver of the differences in the CPC-based and Stage

IV-based results in the left panel of Fig. 5. The following subsections explore the hydrologic processes that are embedded within these process-based flood frequency curves.

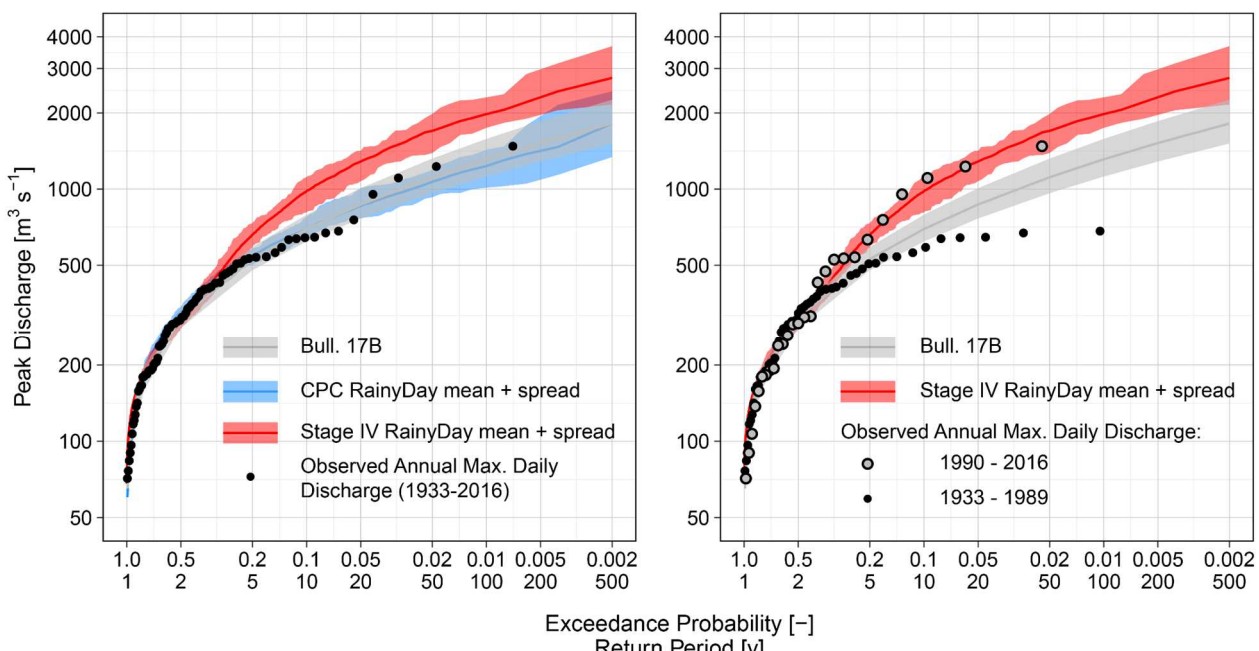

**Figure 5.** Three peak discharge analyses for Turkey River at Garber, IA: RainyDay with Stage IV (2002-2016) and CPC-(1948-2016) rainfall and USGS frequency analyses using Bulletin 17B methods. Shaded areas denote the ensemble spread (RainyDay-based results) and the 90%

confidence intervals (Bulletin 17B-based analysis), respectively. All observed annual maxima daily streamflow from 1933 to 2016 are shown in one group in the left panel, but are separated into two groups in the right panel. Stage IV and Bulletin 17B curves are identical in the two panels.



## 5.3 Simulated Flood Seasonality

As shown in Sect. 5.1, the recent climatology of flooding in Turkey River watershed shows peak flood activity during March-April, with elevated activity continuing through July, reflective of the regional flood "mixture distribution" (e.g. Smith et al., 2011). March-April flooding is associated with springtime rains, high soil moisture, and potentially snowmelt processes, while

May-July flooding results from warm-season organized thunderstorm systems. It is important that any process-based FFA approach capture the influence of this mixture on the flood frequency curve.

The seasonal distribution of simulated flood occurrence and magnitude using Stage IV- and CPC-based results show that most simulated floods in our process-based approach occur between March and June (Fig. 6), in accordance with observed annual maxima daily discharge (Fig. 3c; see also Fig. 7b). The peak of occurrence using Stage IV is shifted several weeks later than

the CPC-based results, which agrees with the recent shift in seasonality of flood observations shown in the Fig. 3c. Although most simulated events still occur around April, our results show that the largest peaks occur later, in May-September. This is consistent with Villarini et al. (2011), who showed that summertime organized convective systems are responsible for some of the largest peaks in Iowa.

Figure 6 shows that rainfall events around August have the potential to cause severe flooding, despite the lack of observed

August flood peaks in Turkey River. The Stage IV- and CPC-based storm catalogs, which includes major storms from the surrounding region, includes large late-summer storm events, which produce large floods within the process-based analysis. This suggests that the general lack of major late-summer floods in the observational records for Turkey River may not be a feature of the "true" (unknown) distribution of flooding in the watershed, but rather of undersampling of this distribution in the observed flood record. This result is supported by analysis or regional flood observations (Villarini et al. 2011), and points

to the potential for SST to improve representations of seasonal variations in extreme rainfall relative to local observations.



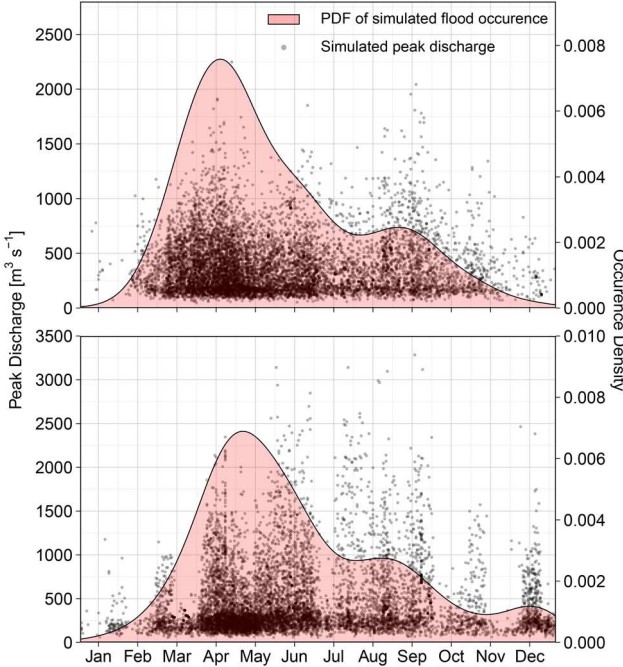

**Figure 6.** Time of occurrence during the year for simulated peak discharge in Turkey River at Garber.

We also examined how hydrological model process representation influence FFA results. We showed previously that the HBV snowpack routine produces trends in simulated daily streamflow that are less realistic than simulated trends without the routine

5 (Fig. 4c, 4d). Interestingly, however, we found very similar flood frequency curves regardless of whether the snowpack routine is used (Fig. 7a) despite very different simulated seasonality (Fig. 7b). With snowpack routine, our approach simulates many large floods (over 1000 m³/s) between February and April (Fig. 7b). This is due to high March-April soil moisture value (Fig. 7c, 7d) associated with snowmelt, which increases the probability of flood occurrence during this period. The flood seasonalities derived from historical observations and from the simulated results without the snowpack module, meanwhile,

10 do not exhibit the very frequent April floods that are present in the simulations with the snowpack routine. This example shows that process-based frequency analyses can be subject to issues related poor hydrological model process representation which can produce "correct" results for the wrong reasons. This implies that the modeler must either have sufficient data to diagnose such issues (as we have done here) or have sufficient prior knowledge of the seasonally varying flood processes in her study area to recognize such pitfalls.





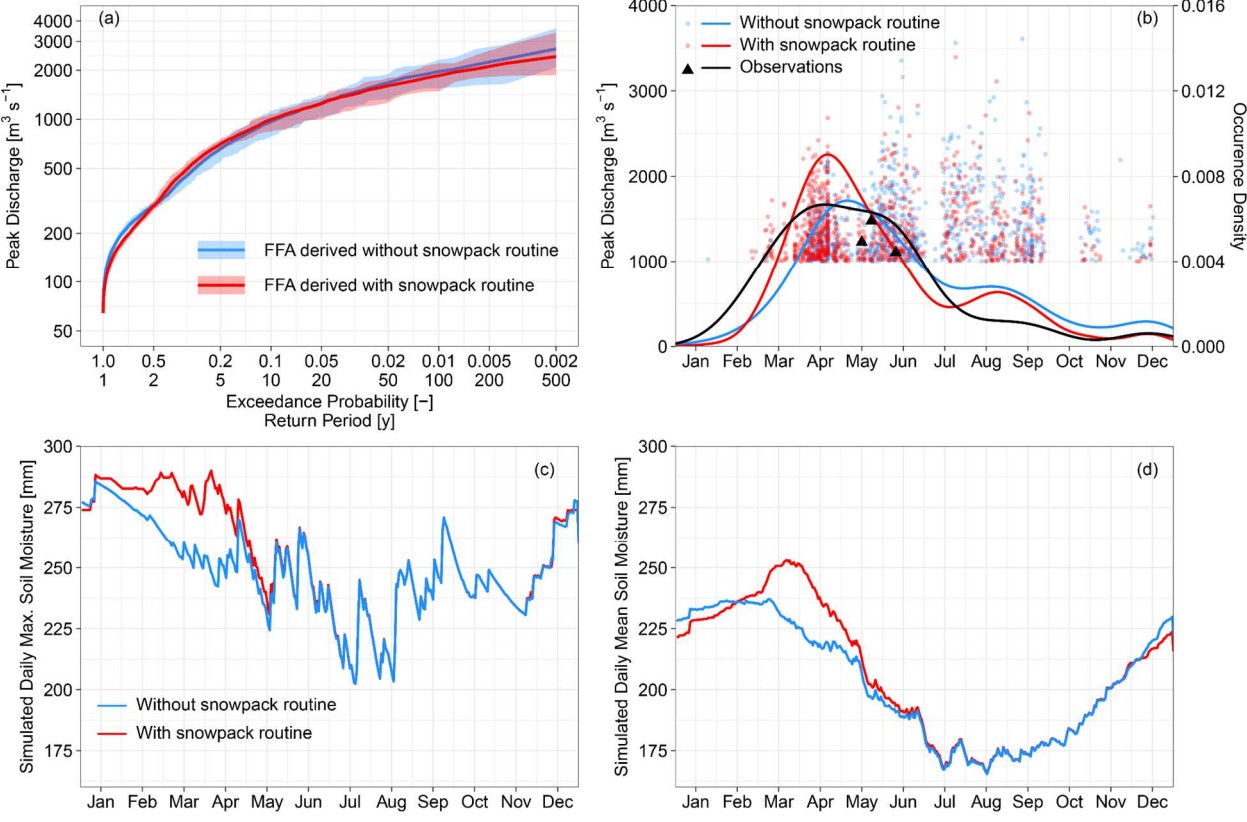

**Figure 7.** Flood frequency curves **(a)** and simulated floods seasonality **(b)** derived by RainyDay-based approach with Stage IV precipitation with (red) or without (blue) the HBV snowpack routine. Simulated discharge larger than 1000 m³/s (dots) and observed peak streamflow (1990-2016) larger than 1000 m³/s (triangles) are shown; discharges below 1000 m³/s are omitted for clarity. The maximum **(c)** and mean **(d)** of simulated soil moisture for each day of the year are shown.

The results shown in Fig. 7 also illustrate a key issue in FFA using both statistical approaches and process-based methods: flood quantiles, though the product of physical processes, reveal little about the underlying processes. This is particularly problematic in changing hydroclimatic or watershed conditions, because nonstationary behavior is likely the result of seasonal shifts in one or more processes. Failure to recognize shifts could lead to incorrect predictions of future conditions. For example, our findings using the HBV snowpack routine predict that most floods are due to high springtime soil moisture due to snowmelt (Fig. 7c, 7d). If we were to project future flood frequency in a warming climate, we might conclude that these spring floods will diminish in importance and thus the tail of the flood distribution will decrease in magnitude. Observations, in contrast, show that an important shift toward summertime flooding has occurred, which may imply the opposite behavior in the tail of the flood distribution in Turkey River since warm season convective rainfall extremes are predicted to increase (e.g. Prein et al., 2016).




### 5.4 Comparison between rainfall and peak discharge quantiles

We examined the relationships between the return periods of 96-hour basin-averaged rainfall accumulations and simulated peak discharge for Turkey River at Garber using Stage IV-based results (Fig. 8; CPC-based results show similar patterns and thus are not shown here). Antecedent soil moisture for each simulated event is shown. Similar to Wright et al. (2014a), Fig. 8

shows that simulated peak discharge quantiles can differ substantially from the rainfall quantiles of the rainfall that produce them. For instance, 500-year ($p_e = 0.002$) rainfall events can cause simulated peak discharges ranging from 21-year ($p_e = 0.048$) to 500-year ($p_e = 0.002$), corresponding to a range in peak discharge of 890 to 2990 m³/s. Results indicate that the peak discharge quantiles are always larger than the rainfall quantiles in wet antecedent soil moisture conditions, while the reverse is true in for dry conditions. These results also demonstrate that the DS assumption of 1:1 equivalency between rainfall

and peak discharge quantiles does not hold in Turkey River. Rainfall spatial variability and drainage network structure, which are ignored in this study due to the lumped nature of HBV, further complicate the relationship between rainfall and discharge quantiles.

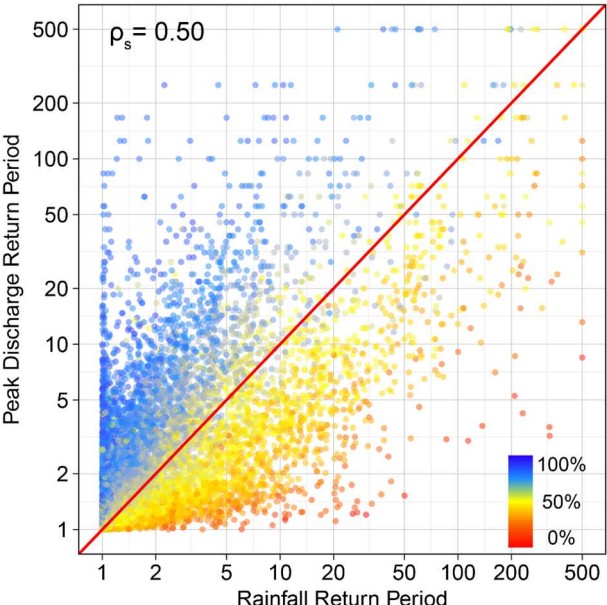

**Figure 8.** Relationships between rainfall return periods and simulated peak discharge return periods estimated via RainyDay (process)-based
method using Stage IV rainfall data. Spearman rank correlation $\rho_s$ is given. Shading color indicates the normalized modeled antecedent soil moisture value, which is calculated as $Normalized\ soil\ moisture = \frac{soil\ moisutre - min.soil\ moisutre}{field\ capacity - min.soil\ moisutre} * 100\%$.

We further examine the relationship between annual rainfall and annual flood peak maxima. In Sect. 2.2, we pointed out that DS methods utilize IDF curves, which are usually estimated using annual maxima from rain gage records, to estimate quantiles from the distribution of annual discharge peaks. In our process-based FFA approach, however, we do not assume that annual

discharge maxima are the result of the largest rainfall event of the year. Rather, lower-magnitude rainfall events, combined with high soil moisture, could produce the highest discharge. Table 2 shows the percentage of annual peak flow driven by



annual maximum gains with increasing return period for both CPC-based and Stage IV-based results. For simulated peak flow with $p_e > 0.01$, a large portion of simulated annual peak flow is not caused by the annual maximum rainfall. For rarer peak flows ($p_e \leq 0.01$), over 90% of these flood events are driven by the annual maximum rainfall, pointing to the fact that the tail of flood peaks is driven by extreme rainfall, with antecedent conditions playing a modulating role.

**Table 2.** Percentages of simulated annual peak flow driven by annual maximum 96-hour rainfall.

| Return Period | Driven by Annual Maximum Rainfall | |
| --- | --- | --- |
| | CPC-based results | Stage IV-based results |
| 1-2 | 29% | 36% |
| 2-5 | 46% | 48% |
| 5-10 | 59% | 66% |
| 10-20 | 70% | 75% |
| 20-50 | 76% | 80% |
| 50-100 | 81% | 88% |
| 100-200 | 90% | 97% |
| 200-500 | 98% | 93% |

## 6 Summary and Conclusions

Interactions between rainfall, land cover, river channel morphology, and watershed antecedent conditions are important drivers of flood response. Standard approaches to estimate extreme flood quantiles (termed flood frequency analysis; FFA), however, take a superficial view of these interactions, as argued in Sect. 2. This study presents an alternative FFA framework that

combines elements of observational analysis, stochastic rainfall generation, and continuous and event-scale rainfall-runoff simulation. We apply the framework to Turkey River, an agricultural watershed in the Midwestern United States that is undergoing significant hydroclimatologic and hydrologic changes which is increasing the magnitude of the largest flood events and shifting their occurrence from the spring to summer. The aim is to estimate flood quantiles by reconstructing meteorological and hydrological processes and their interactions. Unlike FFA using statistical analysis of discharge

observations, such a "process-based" approach is well-suited to nonstationary environments (see also Sivapalan and Samuel, 2009). A key innovation of our FFA approach is that it explicitly considers flood seasonality, which can be an important component of nonstationarity.

We use Stochastic Storm Transposition (SST) to create and resample from "storm catalogs" developed from both 15 years of high-resolution bias-corrected radar rainfall dataset and 69 years of gridded rain gage observations to produce large numbers

of rainfall scenarios for Turkey River. Unlike design storm approaches to FFA, the synthetic rainfall scenarios derived by the SST-based procedure do not require any assumptions regarding the spatial and temporal structure of rainfall, since they are driven by the structure and variability of historical observed storms. Unlike discharge-based statistical analyses, our approach

helps shed light on the physical processes that shape flood frequency. Resampling and spatial transposing of observed rainstorms from the surrounding region makes it feasible to generate extreme precipitation scenarios using relatively short rainfall records. In nonstationary rainfall conditions, recent rainfall data can produce more realistic rainfall scenarios and flood quantile estimates than methods that rely on longer records.

Our analyses show that using the most recent 15 years of rainfall can produce realistic "present-day" flood quantile estimates. Use of longer records, both within our procedure and conventional statistical FFA methods, underestimates current flood frequency due to their inability to represent recent shifts in flood activity in Turkey River. Our results challenge some common FFA assumptions, including the design storm presumption that rainfall annual maxima produce discharge annual maxima and the assumption of 1:1 equivalence in rainfall and flood quantiles. We paint a more complex picture in Turkey River, in which

the shifting seasonality in rainfall and watershed conditions combine to shape the flood frequency.

Spatial variability in rainfall structure, soil moisture, land use and watershed morphology, which are ignored in this study due to the use of a lumped hydrological model, as further complexity to the flood generating processes. However, the proposed framework can be employed with more sophisticated distributed hydrological models, thus facilitating the examination of rainfall spatial variability and its interactions with other factors (e.g. heterogeneous watershed characteristics and river network

processes). This coupling may prove particularly useful for FFA in large watersheds in which there is a practically infinite number of different combinations of such factors that could produce floods—a population that is almost certain to be undersampled in stream gage records and poorly served by design storm assumptions.

Our framework highlights the opportunity and challenge with process-based FFA approaches; namely, that progress on understanding and estimating flood frequency and how it is evolving in an era of unprecedented changes in land use and

climate requires better understanding of how the underlying physical processes, and the interactions between them, are changing. Poor model representation of key hydrological processes, however, can lead to incorrect conclusions about present or future flood frequency. Despite the challenge, we share the view of Sivapalan and Samuel (2009), however, that process-based approaches hold great potential for advances in FFA research and practice.

**Software and model code**

The RainyDay software is available at Github container (https://github.com/danielbwright/RainyDay2) and a web-based version is available at Daniel. W's research group website (http://her.cee.wisc.edu/projects/rainyday).

**Competing interests**

The authors declare that they have no conflict of interest.



## Acknowledgements

Guo. Y's and Daniel. W's contributions were supported by the U.S. National Science Foundation (NSF) Hydrologic Sciences Program (award number 1749638) and by the Bureau of Reclamation Research and Development Office Project ID 1735, which also supported Kathleen. H's contributions. Cassia. S's contributions were supported by the U.S. National Aeronautics and Space Administration's MUREP Institutional Research Opportunity. Zhihua. Z's contributions were supported by Sun Yat-sen University. This study used computing resources and assistance from the UW-Madison Center For High Throughput Computing (CHTC), which is supported by UW-Madison, the Advanced Computing Initiative, the Wisconsin Alumni Research Foundation, the Wisconsin Institutes for Discovery, and NSF, and is an active member of the Open Science Grid, which is supported by NSF and the U.S. Department of Energy. We thank M. Booij and M. Vis for their guidance on the HBV model.

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
