# Peer review of "Process-Based Flood Frequency Analysis in an Agricultural Watershed Exhibiting Nonstationary Flood Seasonality"

_Hydrology and Earth System Sciences, 2018_

## Referee Comment (RC1) · Anonymous Referee #1 · 22 Nov 2018

Comments on "Process-based flood frequency analysis in an agricultural watershed exhibiting nonstationary flood seasonality", by Yu et al., submitted to HESS.

The authors explore the utility of hydrological simulations driven by stochastically transposed rainfall fields in deriving flood frequency over a watershed that experiences nonstationarities. Their results highlight the importance of considering changing flood seasonality in flood frequency analysis. While process-based approaches have a fair amount of advantages, their shortcomings are also quite obvious, for instance, mode uncertainty in both parameters and model structure, representation of synthetic rainfall scenarios, etc. As a hydrologist, I would still favor statistical approaches if the gaug-

ing record is good (as is the case in this paper). This being said, I would suggest the authors focus on explaining the importance of changing flood seasonality in flood frequency, but rather demonstrating the superiority of process-based approaches to other FFA methods (which is not, as far as I can see). My specific comments are listed below.

Specific comments:

1. An important part is missing from the present paper is model validation. Evidence needs to be explicitly presented to show the capability of long-term model simulations in capturing, for instance, flood seasonality, as well as other features (distribution of annual maximum discharge). This can be done by adding simulation results into Figure 3b and Figure 5a. The authors show a larger frequency of floods during post-summer season in their simulations, could this be possibly related to the positive model biases in representing rainfall-runoff processes during this season? The reliability of process-based approaches in FFA builds on decent model simulations. The authors should spend additional efforts in demonstrating this in the paper. This can be done by providing a quantitative assessment of the model performance. Another question about the simulation, how is channel flow represented/considered in the analyses. Antecedent streamflow in the channels can be an important element in representing antecedent watershed wetness, in addition to soil moisture, that plays a role in streamflow simulation.

2. The representation of synthetic rainfall fields is another key in process-based FFA approaches. The authors mentioned that they chose 'most intense rainfall events' within a prescribed domain. How exactly do they define "most intense rainfall events"? Please explain. The authors use the word "realistic" throughout the paper which is inappropriate or miss-leading. They are using synthetic rainfall fields, even though based on real storm events. Please modify.

3. The authors show flood frequency estimates in modern times using Stage IV rainfall

fields, and the results match well with gauging records. How about the performance of CPC rainfall in estimating flood frequency?

4. An interesting finding in the paper is described in P17 Line 15-20, but needs to be rephrased. We can see summer floods dominate the upper tail of flood frequency in this region, even though they do not occur as frequent as spring floods. The distribution derived from gauging records is still the 'truth' anyway. Under-representation of summer floods is a pretty common feature of flood peak distributions in the US. I would suggest the authors to provide a brief diagnostic summary of the most extreme flood events in this region.

5. The authors compared simulation results using model with and without snow module, and suggest in the paper that "the modeler must either have sufficient data to diagnose such issues or have sufficient prior knowledge." (P18 Line 14). I would believe a snow module should be needed in simulation hydrological regimes in this region (dominant spring floods in flood frequency). We cannot simply opt out the snow module by simply checking the simulation. What prior knowledge do the authors have? I would suggest the authors to examine the observed snow climatology over this region, and more ideally, carry out detailed diagnostic analyses of flood agents in this region.

6. P22 Line5-7, it is not true that conventional statistical FFA methods underestimate flood frequency. At this stage, I would still believe statistical estimates are the ground truth, which enables the evaluation of the process-based approach. The authors do not show updated Bulletin 17B curves using the 1990-2016 flood records in Figure 5, which I would suggest to update. As I have mentioned earlier in general comments, it is not wise for the authors to demonstrate the dominating superiority of process-based FFA approaches in this paper, at least for this region. Process-based approach, as presented in this paper (hydrological model + SST), can be highly recommended in poorly gauged watersheds. For poorly-gauged watersheds, however, another issue arises as how to obtain a large ensemble of antecedent watershed wetness conditions used in event-based model simulations. The authors need to provide a discussion

about both pros and cons of the proposed approach.

I have a couple of additional comments on word expressions, paragraph organizations, etc., but they can wait till the second round of review. The paper can be a worthwhile contribution to the literature subject to major revisions.

———————————————

---

## Referee Comment (RC2) · Anonymous Referee #2 · 6 Jan 2019

The work presents an investigation of flood frequency in the Turkey River basin in the Midwestern United States. The proposed framework, referred to as "process-based" FFA, uses stochastic storm transposition to generate synthetic storms and a lumped hydrologic model to simulate discharge at the outlet of the basin. The authors carry out a series of simulations and corresponding analyses of flood frequency to investigate the impact of seasonality in FFA and potential changes between past and present conditions. Overall, the work has several nice features and the questions posed by the authors are interesting. However, I have some major concerns about certain elements of the proposed framework that need to be addressed before the work can be considered for publication. I provide below major and minor comments that will hopefully help

to improve the clarity and the main findings of this work.

Major comments

1. My first and most important concern about the proposed work is related to the choice of the hydrologic model used. The authors mention in different sections themselves that using a lumped model has several limitations. It is good that they acknowledge this limitation themselves but this does not solve the problem. In fact, based on statements as in Line 13, Page 15 "We did not use the snowpack routine....it was shown to produce unrealistic streamflow results" and given that snow processes are important in the selected basins, one immediately recognizes that the choice of the model is not appropriate. If we combine this with the author's statement in conclusions "L22-23, page 22: Poor model representation of key hydrological processes, however, can lead to incorrect conclusions about present and future flood frequency"…...I am very skeptical about the conclusions derived based on this model's results. If the model cannot represent well snow processes (particularly flooding due to rain on snow, which should be important in the area) then I fear that the "process-based" FFA is flawed. In this case, the work should be presented at most as a sensitivity analysis and statements such as L1,P22 "helps shed light on the physical processes that shape flood frequency" should be rephrased accordingly.

2. The calibration and validation of the model lacks clarity. Which forcing was used to calibrate the model? And how the model was validated? These points are not clear in section 4.1. Then in section 5.2 L13,P15 "Different HBV parameters are used..." suggests that separate parameterization was used for the different precipitation forcing but no evidence is provided on a) the validation of the model for the two dataset and b) the variability in model parameters. For the later, if the parameters are significantly different, it will highlight further problems with the approach since this will mean that CPC-HBV and CPC-Stage IV simulations treat hydrological processes differently (i.e. may give more weight to different processes in each case). This needs to be investigated and clearly explained in order to understand whether the results can be

considered "realistic" or are results of a numerical exercise that mixes two different things.

3. For the results in Fig5 right panel: Do you use soil moisture years prior to 1990 for the StageIV process-based approach? Also, you should apply the Bull. 17B for the two periods (1933-1989 and 1990-2016) and add them on the graph for comparison.

Minor comments

1. P1,L18: "a watershed that is undergoing significant climatic... change". Is the climatic change at the scale of the watershed only? Consider revising.

2. P16L2: "but higher estimates" should be "but gives higher estimates"?

3. Fig.6: Improve caption. What is the upper and what the lower panel?

4. P18L13: "processes in her" should be "processes in his/her"

---

## Referee Comment (RC3) · Anonymous Referee #3 · 11 Jan 2019

General comments:

The authors apply a stochastic rainfall generator to provide input for derived flood frequency analyses (DFFA). The rainfall generator uses the storm transposition technique (SST) with the advantage to trade space for time to compile a large enough sample of rainfall events. They combine a continuous hydrologic simulation using observed rainfall time series with an event based simulation using stochastic rainfall events. The former provides realistic initial conditions for the event based runs with the stochastic data. The authors conclude that a) short rainfall observation period can provide reliable flood frequency estimates using SST for contemporary conditions, that b) the

non-stationarity in seasonality can be handled well and that c) inadequate model representations lead to errors.

This combination of continuous and event based modelling is a quite novel idea and provides a flexible framework for DFFA. The application of the methods seems sound, the research is done systematically and the paper reads quite well. However, I do have some concerns regarding the selection of the hydrological model, the selection of two precipitation data sets and some of the conclusions. I will detail these below in the major comments, followed by some minor comments. The paper is worth to be published after major revision.

Major comments:

1) The selection of the lumped HBV model is not plausible to me, especially given that a) the snow routine is not working and b) the high resolution Stage IV rainfall data cannot be utilized by this lumped model.

2) The application of two rainfall data sets is not plausible and also quite confusing for the reader since a) the Stage IV rainfall data observation period (2002-2016) is covered also by the CPC rainfall data observation period (1948-2016), b) a lumped hydrological model cannot really benefit from high resolution rainfall data (see 1) and c) the hydrological simulation results for both rainfall data sets seem to be very similar (as the authors state on page 16, lines 12-13). I would recommend to do all the simulations with the CPC rainfall if the hydrological model is not changed. If a more suitable hydrological model is selected the two data sets might be kept in the study but the differences in hydrological response using the two data sets for the same time period (2002-2016) need also to be demonstrated and discussed.

3) The application of a model without snow routine for a catchment with significant snow processes doesn't make sense to me. This way the advantage of process based flood frequency analysis (FFA) is partly lost; obtaining the correct hydrological response for the wrong reason is not satisfying. I am not convinced that the non-stationarity in

seasonality is only due to changed soil moisture conditions from rainfall. Temporarily shifted snow dynamics might play a role as well.

4) I would be careful with the conclusion, that only with this DFFA method non-stationarity in seasonality can be handled well. Also, non-stationary seasonal FFA approaches are available employing mixed distributions for getting final design values. This needs to be briefly discussed.

5) This combination of continuous and event based modelling is a good idea. However, there is an important limitation which should at least be mentioned. The framework provides only one possible realisation of initial conditions. Nature is more variable. Stochastic rainfall models producing continuous rainfall don't pose this limitation on hydrology.

Minor comments:

1. Page 2, line 4: This sentence is confusing. I am assuming you mean '. . .. statistical analysis of observed streamflow, design storms !and! continuous simulation !or! other so called "derived" or "process based" methods.'

2. Page 4, lines 15-17: This sentence seems not to be complete.

3. Page 10, steps 3 and 4: I would stress that the 30 storms per year are randomly transposed over the domain, only sometimes hitting the catchment and sometimes not. They are not all transposed on the catchment, which would lead to an overestimation of the flood frequency. The reader not familiar with your method might misunderstand that.

4. Page 11, lines 8-9: The selection of the largest event per year for FFA might also be mis-understood. Here, it also needs to be considered that many of the 30 events do not produce any flood if they do not hit the catchment (see comment 3).

5. Page 14: line 2: Should it not be ". . . but overestimates for pe<0.3 . . ."
6. Fig. 5: Why did you select the period 1990 – 2016 and not 1980 or 1970 as starting year? This needs to be justified.

7. Fig. 5: I would also add a statistical analysis (Bull 17.b) for the contemporary period (1990-2016) for comparison.

8. Fig. 6: There is no description neither in legend nor in figure caption about the source of the two figures. I assume they stem from different precipitation data sets?

---

## Author Comment (AC1) · 14 Feb 2019

*Replies to the comments of Anonymous Referee #1*

Responses are provided in blue and proposed revision are in Red. Original reviewer comments are in black. Line and page numbers refer to the original manuscript.

Based upon comments from all three reviewers, we have revisited our model calibration procedure and have been able to obtain acceptable performance from the snowpack routine. This involved a "2-step" calibration process in which warm season processes are calibrated first, and then "warm season parameters " are held constant during subsequent calibration of snowpack-related parameters. This recalibration of HBV is done using both CPC and Stage IV rainfall. We have also added a section on model validation to the revised manuscript, again based on comments from all three reviewers requesting additional validation results. Since all three reviewers provided critiques on these topics, we discuss these two changes before addressing specific comments from individual reviewers.

We have revised model calibration part in the original manuscript, P9, line 15-24, to:

We calibrated the HBV models using both CPC and Stage IV rainfall, and most parameters are the same for CPC- and Stage IV-based models, except for three snow routine parameters (TT, CFMAX, SFCF) and three recession coefficients (K0, K1, K2), allowing for the variability of model parameters for different climate conditions. For each model setup, we first calibrated the model with snowpack routine "turned off" (by setting TT parameter to a very low value) to obtain parameters that can simulate summer floods adequately. Then, keeping these optimized non-snow routine parameters unchanged, we calibrated the snow routine parameters.

To determine the optimized model parameter sets in each procedures, we followed the Genetic Algorithm and Powell (GAP) optimization method as presented by Seibert (2000), which is briefly summarized here. First, 5000 parameter sets are randomly generated from a uniform distribution of the values of each parameter (Table 1), which were then applied to the HBV model in order to maximize Kling Gupta Efficiency (Gupta et al., 2009) of simulated daily discharge. After the GAP has finished, the optimized parameter set were fine-tuned using Powell's quadratic convergent method (Press, 1996) with 1000 additional runs. Lastly, the optimized parameter set was manually adjusted to improve the fits between observed and simulated annual peak flow (see Lamb, 1999). More elaborate calibration and uncertainty estimation procedures such as Generalized Likelihood Uncertainty Estimation (GLUE; Beven and Binley, 1992; Beven, 1993; Beven and Binley, 2014) could be used, but are outside the scope of our study.

After calibration, HBV (two different parameter sets) was used to perform CS with historical CPC and Stage IV rainfall and temperature data to derive long-term simulated soil moisture and snowpack values, which are usually difficult to obtain via measurement. We "pair" samples of these initial conditions with synthetic rainfall events, as described in Sect. 4.2 and Sect. 4.3.

**Table 1.** Overview of HBV model parameters and prior parameter boundaries.

| Parameter | Description | Units | Min value | Max value |
|---|---|---|---|---|
| Snow Routine | | | | |
| TT | Threshold temperature for liquid and solid precipitation | °C | -3 | 3 |
| CFMAX | Degree-day factor | mm d$^{-1}$°C$^{-1}$ | 0.5 | 4 |
| SFCF | Snowfall correction factor | - | 0.5 | 1.2 |
| CFR | Refreezing coefficient | - | 0.01 | 0.1 |
| CWH | Water holding capacity of the snow storage | - | 0.1 | 0.3 |
| Soil Moisture Routine | | | | |
| FC | Maximum soil moisture storage (field capacity) | mm | 100 | 550 |
| LP | Relative soil water storage below which AET is reduced linearly | - | 0.3 | 1 |
| BETA | Exponential factor for runoff generation | - | 1 | 5 |
| Response Routine | | | | |
| PERC | Maximum percolation from upper to lower groundwater box | mm d$^{-1}$ | 0 | 10 |
| UZL | Threshold of upper groundwater box | mm | 0 | 50 |
| K0 | Recession coefficient 0 | d$^{-1}$ | 0.5 | 0.9 |
| K1 | Recession coefficient 1 | d$^{-1}$ | 0.15 | 0.5 |
| K2 | Recession coefficient 2 | d$^{-1}$ | 0.01 | 0.15 |
| Routing Routine | | | | |
| MAXBAS | Length of triangular weighting function | d | 1 | 2.5 |

We have also added "Section 5.2 Model Validation" by modifying the original paper, P13-14, to:

**5.2 Model Validation**

We validated the performance of HBV continuous simulation with respect to flood seasonality, frequency of annual daily discharge maxima, and normalized peak flow (i.e. the simulated or observed daily discharge divided by the 2-year flood), using both Stage IV and CPC as precipitation inputs (Fig. 4). We also validated two structures: one with and the other without the HBV snowpack module. The purpose for this latter validation effort is to highlight the importance of proper process representation (and subsequent validation) in process-based FFA.

Simulated flood seasonality varies substantially during the CPC period of record (1948-2016) depending on the inclusion of the snowpack routine. Differences are less for the Stage IV period of record (2002-2016), due to the decreasing role of snowpack in deriving the floods in recent years (Fig. 4a). In both cases, the seasonality of flooding simulated using HBV is improved with the inclusion of the snowpack module, with a higher (lower) frequency of springtime (summertime) floods which more closely resembles observations. Empirical (i.e. plotting position-based) distributions for the simulated annual daily discharge maxima are mostly within the 90% confidence interval (obtained by nonparametric bootstrap) of the observations (Fig.

4b). The CPC-based simulations differ considerably depending on the inclusion of the snowpack module for more common events, but differences in simulated maxima vanish as flood magnitude increases (e.g. AEP<0.1). This is because the most extreme flood events occur later in the season and are thus independent of snowpack or snowmelt processes. Differences are generally negligible between Stage IV-based simulations with and without snowpack, since floods in this shorter, more recent period are generally driven by summertime thunderstorms. These findings are consistent with the general understanding of the regional seasonality of flooding in the region, as discussed in Sect. 5.1.

We compared all simulated and observed flood peaks that can be associated with a USGS observed daily streamflow value that is at least three times the mean annual daily discharge (Fig. 4c). When associating simulated and observed flood peaks, we look within a 2-day window to allow for modest errors in simulated flood peak timing. All peaks in Fig. 4c are normalized by the median annual (i.e. 2-year) flood, which, as a rule of thumb, can be considered as the "within bank" threshold. Again, HBV with the snowpack routine outperforms the model without it, especially for the small to modest flood events in CPC-based simulations. The model without snowpack routine underestimate the small to modest flood events in two cases due to the neglect of water flux from potential snowmelt. While modest scatter exists in the Stage IV-based simulated peaks, there is no obvious systematic bias with event magnitude when the snowmelt routine is included.

[Figure]

**Figure 1.** HBV model validation for flood seasonality **(a)**, frequency of annual max. daily discharge **(b)** and normalized peak flow (c). For each panel, the corresponding model validation is performed against CPC- (1948-2016) and StageIV-based (2002-2016) simulation and the results derived from HBV model with (without) snowpack routine are shown in blue (red). The 90% confidence interval for observed max. daily discharge (empirical distribution) is derived using the bootstrapping approach. Peak discharge is defined as a data point with USGS observed value that is at least three times the average observations, and peak discharge are normalized by the median of annual daily discharge maxima (i.e. the 2-year flood). Straight black lines indicate 1:1 correspondence, while dashed lines indicate the envelope within which the modeled values are within 50% of observed.

We also validate HBV's snowpack routine using observed GHCN daily snow depth for two simulation periods (Fig. 5a, 5b) and using USGS daily streamflow observations for Stage IV-based period (Fig. 5c). Because of their differing spatial resolutions and physical representations, point-scale GHCN daily snow

depths cannot be directly or quantitatively compared to the watershed-scale snow water equivalent simulated by HBV. Therefore, we validate the snowpack simulation in terms of the snowpack occurrence, defined as the number of occurrences where snow is present on a particular date divided by the total number of years in the historical record. For example, there are 50 days where snowpack is present on January 1st in the 69-year period from 1948-2016, based on GHCN observations and thus the corresponding occurrence rate is 0.72 (50 divided by 69). The HBV model with the snowpack routine captures the central tendency of observed snowpack dynamics, showing that snowpack frequently exists from early November to mid-February, with frequency of snow decreasing from late February until disappearing in early April.

[Figure]

**Figure 2.** The comparison of percent of days with snowpack present between observations and simulations (a, b) and hydrograph validation for StageIV-based simulation (c). For each day within a year, the percent of snowpack existing days is calculated as the ratio of the number of years when snowpack is present to the total years (69 years for CPC and 15 years for StageIV). Observed and simulated hydrograph are normalized by the median annual flood, which is indicated by the dashed blue line.

Model hydrograph validation is provided in Fig. 5c for the Stage IV period (2002-2016), when major flooding occurred throughout Iowa. Model performance shows no obvious evidence of systematic bias in the streamflow simulations. Although flood seasonality derived by Stage IV-based simulation differs slightly from observations (Fig. 4b), these mismatches are associated with flood events smaller than the median annual flood (blue dash line in Fig. 5c). Stage IV-based simulations do not show bias flood magnitude in late summer. In other words, remaining biases in terms of flood seasonality generally

correspond with frequent, small-magnitude events that are typically of less interest in FFA. We therefore conclude that the HBV model with snowpack is generally suitable for subsequent process-based FFA.

The authors explore the utility of hydrological simulations driven by stochastically transposed rainfall fields in deriving flood frequency over a watershed that experiences nonstationarities. Their results highlight the importance of considering changing flood seasonality in flood frequency analysis. While process-based approaches have a fair amount of advantages, their shortcomings are also quite obvious, for instance, mode uncertainty in both parameters and model structure, representation of synthetic rainfall scenarios, etc. As a hydrologist, I would still favor statistical approaches if the gauging record is good (as is the case in this paper). This being said, I would suggest the authors focus on explaining the importance of changing flood seasonality in flood frequency, but rather demonstrating the superiority of process-based approaches to other FFA methods (which is not, as far as I can see).

We thank the reviewer for these useful critiques, that have been very helpful in improving the paper. We fully agree that, particularly in situations of plentiful stream gage observations, statistical approaches are generally preferable. It was never our intention to suggest that our approach is superior to such methods. We note in the original manuscript (P4, line 23-26), however, that there have been prior studies that have demonstrated situations in which rainfall-runoff modeling approaches of various kinds can outperform statistical methods. This, combined with the relative immaturity of rainfall-runoff model-based FFA approaches compared with statistical methods, suggests that additional research, of the kind we present here, can and should be done. As the reviewer stresses, one of the things that such research can point to is the importance of processes and their changes (e.g. seasonal to interannual). In our revised manuscript, we have attempted to emphasize our viewpoint on these issues more clearly. Example revisions to this effect include:

- Include the snowpack routine in the HBV model for both CPC- and StageIV-based simulations.
- Modify the model calibration part (Chapter 4.1) in the original manuscript.
- Add a new section for model validation.
- Address the importance of changing flood seasonality in flood frequency.

We have analyzed two sets of CPC-based results, one for 1948-2016 and the other for 2002-2016 to demonstrate how the changes in flood agents affect the FFA results. We have added the following part to Sect.5.3, P17, line 21 of the original manuscript.

To demonstrate that the discrepancies between the process-based FFA results generated using CPC and using StageIV are driven by changes in flood agents, rather than by differences in model structure (i.e. parameter values), we compared FFA results generated using CPC-based for 1948-2016 and 2002-2016, in terms of event rainfall, initial soil moisture, flood type and peak magnitude (Fig. 8). From 2002-2016 (Fig. 8b), there are fewer flood events driven by snowmelt or rain-on-snow but more driven by rainfall, particularly large magnitude flood events (over 1000 m3/s). In addition, some of the rainfall driven floods (upper left of Fig. 8b) from 2002-2016 indicates high initial soil moisture, which are in accordance with the

significant increasing trend of annual precipitation (Table 2). In general, changes in individual flood agents and their interactions can affect flood frequency. Process-based approaches can help illuminate these changes.

[Figure]

Figure 8. The simulated flood magnitude using CPC rainfall during 1948-2016 (a) and 2002-2016 (b) period, and corresponding antecedent conditions sampled from the continuous simulation. The blue triangles represent the snow related flood events (e.g. snowmelt or rain on snow) and grey dots represents the non-snow related flood events (e.g. rainfall driven). The size of the triangles or dots indicate the antecedent soil moisture with higher value in larger shape. The black dash line indicates the 1000m3/s flood magnitudes.

Specific comments 1-1: An important part is missing from the present paper is model validation. Evidence needs to be explicitly presented to show the capability of long-term model simulations in capturing, for instance, flood seasonality, as well as other features (distribution of annual maximum discharge). This can be done by adding simulation results into Figure 3b and Figure 5a.

We thank the reviewer for this suggestion, which was also voiced by the other reviewers. We have include the model validation, as shown at the beginning of this response, to further demonstrate the capability of long-term simulation in capturing the flood seasonality, high flow magnitude and distribution of annual maximum discharge. We hope the reviewers find it to be more convincing that the limited validation that we included in the original manuscript.

Specific comments 1-2: The authors show a larger frequency of floods during post-summer season in their simulations, could this be possibly related to the positive model biases in representing rainfall-runoff processes during this season? The reliability of process-based approaches in FFA builds on decent model simulations. The authors should spend additional efforts in demonstrating this in the paper. This can be done by providing a quantitative assessment of the model performance.

The hydrograph validation plot (Figure. 5b), along with the flood seasonality validation plot (Figure. 4a) shows that the HBV model with snowpack routine can capture the observed flood seasonality and daily streamflow in the long-term simulation. Although model simulates more flood events in late summer (August-September), it is not biased in terms of late summer flood magnitude. Therefore, we believe these simulated extreme late summer flood events (over 1500 m³/s) are associated with the regional late-summer storm events in Iowa, rather than model bias.

Specific comments 1-3: Another question about the simulation, how is channel flow represented/considered in the analyses. Antecedent streamflow in the channels can be an important element in representing antecedent watershed wetness, in addition to soil moisture, that plays a role in streamflow simulation.

The reviewer is correct in general that this should be considered within our framework. The HBV model, however, does not need to sample channel flow (streamflow) for the antecedent conditions. The following equations show how the HBV model calculates the streamflow.

$$Q[t] = Q_0[t] + Q_1[t] + Q_2[t]$$

$$Q_0[t] = K_0 * MAX(SUZ[t-1] + recharge[t] + excess[t] - UZL, 0)$$

$$Q_1[t] = K_1 * (SUZ[t-1] + recharge[t] + excess[t] - UZL - PERC)$$

$$Q_2[t] = K_2 * (SLZ[t-1] + PERC)$$

Where conceptually,

$Q[t]$ is the current time streamflow

$Q_0[t], Q_1[t], Q_2[t]$ are the current time overland flow, intermediate flow and baseflow

$recharge[t], excess[t]$ are the current time flux to groundwater and excess runoff, all of which depend on the soil moisture at previous time step

$SUZ[t-1], SLZ[t-1]$ are the water level in upper and lower groundwater box at the previous time step

$K_0, K_1, K_2, UZL, PERC$ are model parameters

In general, the current time overland flow ($Q_0[t]$) and intermediate flow ($Q_1[t]$) only depend on soil moisture and water level in the upper groundwater box at the previous time step while the current time baseflow $Q_2[t]$ depends on the water level in the lower groundwater box at the previous time step. The more details on HBV model structure can be found in the HBV references in original manuscript, P3, line 4-7.

Specific comments 2-1: The representation of synthetic rainfall fields is another key in process-based FFA approaches. The authors mentioned that they chose 'most intense rainfall events' within a prescribed domain. How exactly do they define "most intense rainfall events"? Please explain.

The RainyDay software selects the most intense rainfall events within the transposition domain, in terms of rainfall accumulation of duration *t* and with the same size, shape, and orientation of

the watershed. For example, the principal axis of the Turkey River watershed in this study is oriented roughly northwest-southeast and has an area of 4002 km$^2$. In this case, the 450 selected storms from the historical rainfall data are those associated with the 450 highest 96-hour rainfall accumulations over an area of 4002 km$^2$ with the same shape and orientation as the Turkey River watershed.

We have modified P10, line 13-14 to:

These intense storms are in terms of 96-hour rainfall accumulation and have the same size, shape, and orientation of the Turkey River watershed, which is oriented roughly northwest-southeast and with an area of 4002 km$^2$. In order to avoid overlapping storms, these selected events must be separated by at least 24 hours.

Specific comments 2-2: The authors use the word "realistic" throughout the paper which is inappropriate or miss-leading. They are using synthetic rainfall fields, even though based on real storm events. Please modify.

We believe that our word choice is reasonable when referring the SST-based rainfall fields. They require no parameterization or assumption regarding their spatial or temporal structure (only their starting location is changed), and thus are objectively more realistic than more conventional stochastic rainfall generators. The "realistic" claim would be admittedly more suspect in an environment with complex terrain features (e.g. mountains, coastlines) where both radar estimates and transposition of rainfall fields would be more suspect. Most references on SST in original manuscript, P9, line 30-31, also used word "realistic rainfall".

Specific comments 3: The authors show flood frequency estimates in modern times using Stage IV rainfall fields, and the results match well with gauging records. How about the performance of CPC rainfall in estimating flood frequency?

The RainyDay based FFA using CPC-Unified rainfall data from 2002 to 2016 closely resembles the Stage IV-based FFA, as we mentioned in original manuscript, P16, line 12-15. Regardless, we have added a supplementary plot showing the CPC, Stage IV and Bull.17B based FFA for the modern time (2002-2016).

Supplementary Fig. 1 shows two features that result using CPC data. First, the extreme tail is underestimated, relative to the Stage IV-based simulations and the statistical approach. CPC is known to contain errors in the extreme tail, due to gage undercatch, insufficient gage density to properly sample convective rain cells, and spatial averaging of such cells over large areas, which effectively reduces peak rainfall depths. Second, CPC overestimates the magnitude of more frequent events. This is likely the result of its coarse spatial resolution, which will "smear" rainfall over larger areas (i.e. entire ~600 km2) grid cells when it should be more localized. This would serve to increase the likelihood of rainfall over the watershed, albeit at relatively lower depths/intensities. Thus, if one is to restrict the time period of the rainfall data to recent years (for example, the 2002-2016 time period for which Stage IV is available), then Stage IV would likely be better. As an aside, this belief that Stage IV is preferable to other precipitation datasets in the United States is widely shared in the satellite precipitation community, where Stage IV is often used as a validation dataset.

[Figure]

**Supplementary Figure 1**. Three peak discharge analyses for Turkey River at Garber, IA: RainyDay with Stage IV (2002-2016) and CPC-(2002-2016) rainfall and USGS frequency analyses (1990-2016) using Bulletin 17B methods. Shaded areas denote the ensemble spread (RainyDay-based results) and the 90% confidence intervals (Bulletin 17B-based analysis), respectively. All observed annual daily streamflow maxima from 1990 to 2016 are shown in black dots.

Specific comments 4: An interesting finding in the paper is described in P17 Line 15-20, but needs to be rephrased. We can see summer floods dominate the upper tail of flood frequency in this region, even though they do not occur as frequent as spring floods. The distribution derived from gauging records is still the 'truth' anyway. Under-representation of summer floods is a pretty common feature of flood peak distributions in the US. I would suggest the authors to provide a brief diagnostic summary of the most extreme flood events in this region.

This is a good suggestion and the newly-added model validation section includes seasonal validations (5.2 Model Validation), as shown at the beginning of this response. Model validations, with respect to flood seasonality, normalized peak flow and hydrograph, show that HBV does not show bias flood magnitude in late summer.

A summary of the most extreme flood events in Iowa are provided in the section 5.1 of the original manuscript, and is provided here: "Flood peak distributions in Iowa "mixtures" of two basic types. Spring floods are associated with springtime rains, high soil moisture, and potentially snowmelt. Summer floods are associated with convective systems. The latter have been shown to significantly affect the upper tail of the flood peak distribution (Villarini et al, 2011) who showed that about 40% of the largest flood peaks are during the May-July period in Iowa. It is important that any process-based FFA approach capture the influence of this mixture on the flood frequency curve." This does not imply that individual gage records are "the truth", only the best representative of it that we have. Thus, discrepancies between model-based approaches and such as ours and observational records warrant further attention.

Specific comments 5: The authors compared simulation results using model with and without snow module, and suggest in the paper that "the modeler must either have sufficient data to diagnose such issues or have sufficient prior knowledge." (P18 Line 14). I would believe a snow module should be needed in simulation hydrological regimes in this region (dominant spring floods in flood frequency). We cannot simply opt out the snow module by simply checking the simulation. What prior knowledge do the authors have? I would suggest the authors to examine the observed snow climatology over this region, and more ideally, carry out detailed diagnostic analyses of flood agents in this region.

This is a very useful critique. We took this advice into account and developed a new calibration approach that avoids some of the pitfalls that we encountered using more standard calibration techniques. As shown above, we validate this new calibration with respect to flood seasonality, hydrograph, normalized peak flow and snowpack. We finally conclude that the snowpack routine of HBV is indeed important in this study region for this application. We appreciate the insistence of all reviewers in this regard, since it has led to a stronger and more defensible methodology.

Specific comments 6-1: P22 Line5-7, it is not true that conventional statistical FFA methods underestimate flood frequency. At this stage, I would still believe statistical estimates are the ground truth, which enables the evaluation of the process-based approach. The authors do not show updated Bulletin 17B curves using the 1990-2016 flood records in Figure 5, which I would suggest to update.

Figure 5 in the original manuscript shows that conventional FFA methods (defined here as usage of stationary statistical distributions fitted to the period of record using a standard fitting software) underestimate flood frequency beyond the 2-5 year recurrence interval. The statistical fits shown in Figure 5 are included to emphasize that we neglect nonstationarity (as is typically done in FFA practice) at our peril, and usage of "old" data in the face of pronounced hydrologic change can produce incorrect results. We therefore must contend that statistical estimates in such situations should not be considered "ground truth." Bulletin 17B-based results using 1990-2016 flood peaks are shown in the figure above (see responses to specific comment 3). This fits the observed flood peaks well, as one would expect, though obviously subject to substantial uncertainty for low AEP events due to the short fitting period. Other methods, such as nonstationary FFA, could be used, but our goal is not to prove the superiority of one method or another, but rather to highlight some important issues regarding flood physical processes, their changes, and the resulting implications for flood frequency, issues which are generally ignored in conventional analysis.

As I have mentioned earlier in general comments, it is not wise for the authors to demonstrate the dominating superiority of process-based FFA approaches in this paper, at least for this region. Process-based approach, as presented in this paper (hydrological model + SST), can be highly recommended in poorly gauged watersheds. For poorly-gauged watersheds, however, another issue arises as how to obtain a large ensemble of antecedent watershed wetness conditions used in event-based model simulations. The authors need to provide a discussion about both pros and cons of the proposed approach.

Again, our intention was not to argue for the superiority of process-based methods, and we regret that we gave the reviewer that impression. We have modified the manuscript to make more clear

the point that we are attempting to highlight the importance of flood processes and their changes in "shaping" flood frequency, and show an approach that can begin to account for such processes and their changes—though more work is needed, and is ongoing within our research group and elsewhere. Additionally, we agree with the reviewer that a brief discussion about both pros and cons of our framework is necessary.

We have revised the last paragraph of the conclusion to:

A number of issues remain that make broader usage of our process-based framework challenging. Perhaps the biggest limitation of process-based approaches is the necessity of discharge observations, which are central to both identifying hydrologic changes and to calibrate and validate the hydrologic model. Thus, usage of the approach in ungaged basins may not produce satisfactory results. This issue is fundamental to other FFA techniques as well. Statistically-based discharge analyses, for example, similarly rely on streamflow observations, while design storm approaches also require hydrologic model calibration.

Our framework highlights the opportunity and challenge with process-based FFA approaches; namely, that progress on understanding and estimating flood frequency and how it is evolving in an era of unprecedented changes in land use and climate requires better understanding of how the underlying physical processes, and the interactions between them, are changing. Poor model representation of key hydrological processes, however, can lead to incorrect conclusions about present or future flood frequency. Despite the challenge, we share the view of Sivapalan and Samuel (2009) that process-based approaches hold great potential for advances in FFA research and practice, particularly in projecting the future FFA when coupled with high resolution climate model. We do not propose that process-based approaches should necessarily supplant more conventional discharge-based analyses, and discharge observations were central to our present study. Rather, we anticipate a gradual "merging" of statistical and process-based stochastic simulation techniques as well as of the associated observations and synthetic data.

I have a couple of additional comments on word expressions, paragraph organizations, etc., but they can wait till the second round of review. The paper can be a worthwhile contribution to the literature subject to major revisions.

We look forward to further feedback from the reviewer. We have also made minor modification to the structure and word choice in the revised version.

---

## Author Comment (AC2) · 14 Feb 2019

Responses are provided in blue and proposed revision are in Red. Original reviewer comments are in black. Line and page numbers refer to the original manuscript.

Based upon comments from all three reviewers, we have revisited our model calibration procedure and have been able to obtain acceptable performance from the snowpack routine. This involved a "2-step" calibration process in which warm season processes are calibrated first, and then "warm season parameters " are held constant during subsequent calibration of snowpack-related parameters. This recalibration of HBV is done using both CPC and Stage IV rainfall. We have also added a section on model validation to the revised manuscript, again based on comments from all three reviewers requesting additional validation results. Since all three reviewers provided critiques on these topics, we discuss these two changes before addressing specific comments from individual reviewers.

We have revised model calibration part in the original manuscript, P9, line 15-24, to:

We calibrated the HBV models using both CPC and Stage IV rainfall, and most parameters are the same for CPC- and Stage IV-based models, except for three snow routine parameters (TT, CFMAX, SFCF) and three recession coefficients (K0, K1, K2), allowing for the variability of model parameters for different climate conditions. For each model setup, we first calibrated the model with snowpack routine "turned off" (by setting TT parameter to a very low value) to obtain parameters that can simulate summer floods adequately. Then, keeping these optimized non-snow routine parameters unchanged, we calibrated the snow routine parameters.

To determine the optimized model parameter sets in each procedures, we followed the Genetic Algorithm and Powell (GAP) optimization method as presented by Seibert (2000), which is briefly summarized here. First, 5000 parameter sets are randomly generated from a uniform distribution of the values of each parameter (Table 1), which were then applied to the HBV model in order to maximize Kling Gupta Efficiency (Gupta et al., 2009) of simulated daily discharge. After the GAP has finished, the optimized parameter set were fine-tuned using Powell's quadratic convergent method (Press, 1996) with 1000 additional runs. Lastly, the optimized parameter set was manually adjusted to improve the fits between observed and simulated annual peak flow (see Lamb, 1999). More elaborate calibration and uncertainty estimation procedures such as Generalized Likelihood Uncertainty Estimation (GLUE; Beven and Binley, 1992; Beven, 1993; Beven and Binley, 2014) could be used, but are outside the scope of our study.

After calibration, HBV (two different parameter sets) was used to perform CS with historical CPC and Stage IV rainfall and temperature data to derive long-term simulated soil moisture and snowpack values, which are usually difficult to obtain via measurement. We "pair" samples of these initial conditions with synthetic rainfall events, as described in Sect. 4.2 and Sect. 4.3.

**Table 1.** Overview of HBV model parameters and prior parameter boundaries.

| Parameter | Description | Units | Min value | Max value |
|---|---|---|---|---|
| Snow Routine | | | | |
| TT | Threshold temperature for liquid and solid precipitation | °C | -3 | 3 |
| CFMAX | Degree-day factor | mm d$^{-1}$°C$^{-1}$ | 0.5 | 4 |
| SFCF | Snowfall correction factor | - | 0.5 | 1.2 |
| CFR | Refreezing coefficient | - | 0.01 | 0.1 |
| CWH | Water holding capacity of the snow storage | - | 0.1 | 0.3 |
| Soil Moisture Routine | | | | |
| FC | Maximum soil moisture storage (field capacity) | mm | 100 | 550 |
| LP | Relative soil water storage below which AET is reduced linearly | - | 0.3 | 1 |
| BETA | Exponential factor for runoff generation | - | 1 | 5 |
| Response Routine | | | | |
| PERC | Maximum percolation from upper to lower groundwater box | mm d$^{-1}$ | 0 | 10 |
| UZL | Threshold of upper groundwater box | mm | 0 | 50 |
| K0 | Recession coefficient 0 | d$^{-1}$ | 0.5 | 0.9 |
| K1 | Recession coefficient 1 | d$^{-1}$ | 0.15 | 0.5 |
| K2 | Recession coefficient 2 | d$^{-1}$ | 0.01 | 0.15 |
| Routing Routine | | | | |
| MAXBAS | Length of triangular weighting function | d | 1 | 2.5 |

We have also added "Section 5.2 Model Validation" by modifying the original paper, P13-14, to:

**5.2 Model Validation**

We validated the performance of HBV continuous simulation with respect to flood seasonality, frequency of annual daily discharge maxima, and normalized peak flow (i.e. the simulated or observed daily discharge divided by the 2-year flood), using both Stage IV and CPC as precipitation inputs (Fig. 4). We also validated two structures: one with and the other without the HBV snowpack module. The purpose for this latter validation effort is to highlight the importance of proper process representation (and subsequent validation) in process-based FFA.

Simulated flood seasonality varies substantially during the CPC period of record (1948-2016) depending on the inclusion of the snowpack routine. Differences are less for the Stage IV period of record (2002-2016), due to the decreasing role of snowpack in deriving the floods in recent years (Fig. 4a). In both cases, the seasonality of flooding simulated using HBV is improved with the inclusion of the snowpack module, with a higher (lower) frequency of springtime (summertime) floods which more closely resembles observations. Empirical (i.e. plotting position-based) distributions for the simulated annual daily discharge maxima are mostly within the 90% confidence interval (obtained by nonparametric bootstrap) of the observations (Fig.

4b). The CPC-based simulations differ considerably depending on the inclusion of the snowpack module for more common events, but differences in simulated maxima vanish as flood magnitude increases (e.g. AEP<0.1). This is because the most extreme flood events occur later in the season and are thus independent of snowpack or snowmelt processes. Differences are generally negligible between Stage IV-based simulations with and without snowpack, since floods in this shorter, more recent period are generally driven by summertime thunderstorms. These findings are consistent with the general understanding of the regional seasonality of flooding in the region, as discussed in Sect. 5.1.

We compared all simulated and observed flood peaks that can be associated with a USGS observed daily streamflow value that is at least three times the mean annual daily discharge (Fig. 4c). When associating simulated and observed flood peaks, we look within a 2-day window to allow for modest errors in simulated flood peak timing. All peaks in Fig. 4c are normalized by the median annual (i.e. 2-year) flood, which, as a rule of thumb, can be considered as the "within bank" threshold. Again, HBV with the snowpack routine outperforms the model without it, especially for the small to modest flood events in CPC-based simulations. The model without snowpack routine underestimate the small to modest flood events in two cases due to the neglect of water flux from potential snowmelt. While modest scatter exists in the Stage IV-based simulated peaks, there is no obvious systematic bias with event magnitude when the snowmelt routine is included.

[Figure]

**Figure 1.** HBV model validation for flood seasonality **(a)**, frequency of annual max. daily discharge **(b)** and normalized peak flow (c). For each panel, the corresponding model validation is performed against CPC- (1948-2016) and StageIV-based (2002-2016) simulation and the results derived from HBV model with (without) snowpack routine are shown in blue (red). The 90% confidence interval for observed max. daily discharge (empirical distribution) is derived using the bootstrapping approach. Peak discharge is defined as a data point with USGS observed value that is at least three times the average observations, and peak discharge are normalized by the median of annual daily discharge maxima (i.e. the 2-year flood). Straight black lines indicate 1:1 correspondence, while dashed lines indicate the envelope within which the modeled values are within 50% of observed.

We also validate HBV's snowpack routine using observed GHCN daily snow depth for two simulation periods (Fig. 5a, 5b) and using USGS daily streamflow observations for Stage IV-based period (Fig. 5c). Because of their differing spatial resolutions and physical representations, point-scale GHCN daily snow

depths cannot be directly or quantitatively compared to the watershed-scale snow water equivalent simulated by HBV. Therefore, we validate the snowpack simulation in terms of the snowpack occurrence, defined as the number of occurrences where snow is present on a particular date divided by the total number of years in the historical record. For example, there are 50 days where snowpack is present on January 1st in the 69-year period from 1948-2016, based on GHCN observations and thus the corresponding occurrence rate is 0.72 (50 divided by 69). The HBV model with the snowpack routine captures the central tendency of observed snowpack dynamics, showing that snowpack frequently exists from early November to mid-February, with frequency of snow decreasing from late February until disappearing in early April.

[Figure]

**Figure 2.** The comparison of percent of days with snowpack present between observations and simulations (a, b) and hydrograph validation for StageIV-based simulation (c). For each day within a year, the percent of snowpack existing days is calculated as the ratio of the number of years when snowpack is present to the total years (69 years for CPC and 15 years for StageIV). Observed and simulated hydrograph are normalized by the median annual flood, which is indicated by the dashed blue line.

Model hydrograph validation is provided in Fig. 5c for the Stage IV period (2002-2016), when major flooding occurred throughout Iowa. Model performance shows no obvious evidence of systematic bias in the streamflow simulations. Although flood seasonality derived by Stage IV-based simulation differs slightly from observations (Fig. 4b), these mismatches are associated with flood events smaller than the median annual flood (blue dash line in Fig. 5c). Stage IV-based simulations do not show bias flood magnitude in late summer. In other words, remaining biases in terms of flood seasonality generally

correspond with frequent, small-magnitude events that are typically of less interest in FFA. We therefore conclude that the HBV model with snowpack is generally suitable for subsequent process-based FFA.

The work presents an investigation of flood frequency in the Turkey River basin in the Midwestern United States. The proposed framework, referred to as "process-based" FFA, uses stochastic storm transposition to generate synthetic storms and a lumped hydrologic model to simulate discharge at the outlet of the basin. The authors carry out a series of simulations and corresponding analyses of flood frequency to investigate the impact of seasonality in FFA and potential changes between past and present conditions. Overall, the work has several nice features and the questions posed by the authors are interesting. However, I have some major concerns about certain elements of the proposed framework that need to be addressed before the work can be considered for publication. I provide below major and minor comments that will hopefully help.

We thank the reviewer for these useful critiques, which have been very helpful in improving the manuscript.

Major comments 1: My first and most important concern about the proposed work is related to the choice of the hydrologic model used. The authors mention in different sections themselves that using a lumped model has several limitations. It is good that they acknowledge this limitation themselves but this does not solve the problem. In fact, based on statements as in Line 13, Page 15 "We did not use the snowpack routine…it was shown to produce unrealistic streamflow results" and given that snow processes are important in the selected basins, one immediately recognizes that the choice of the model is not appropriate. If we combine this with the author's statement in conclusions "L22-23, page 22: Poor model representation of key hydrological processes, however, can lead to incorrect conclusions about present and future flood frequency"…I am very skeptical about the conclusions derived based on this model's results. If the model cannot represent well snow processes (particularly flooding due to rain on snow, which should be important in the area) then I fear that the "process-based" FFA is flawed. In this case, the work should be presented at most as a sensitivity analysis and statements such as L1, P22 "helps shed light on the physical processes that shape flood frequency" should be rephrased accordingly.

This is a valid criticism and we thank the reviewer. We hope that the added model calibration and validation, as shown in the beginning of this response, addresses most of the reviewer's present concern. As shown, we have devised a new calibration approach that provided acceptable performance while included the snowpack routine in the HBV model, since we agree with the reviewer that snow processes are potentially important elements of flooding in the region and should not be omitted.

Major comments 2: The calibration and validation of the model lacks clarity. Which forcing was used to calibrate the model? And how the model was validated? These points are not clear in section 4.1. Then in section 5.2 L13,P15 "Different HBV parameters are used…" suggests that separate parameterization was used for the different precipitation forcing but no evidence is provided on a) the validation of the model for the two dataset and b) the variability in model

parameters. For the later, if the parameters are significantly different, it will highlight further problems with the approach since this will mean that CPC HBV and CPC-Stage IV simulations treat hydrological processes differently (i.e. may give more weight to different processes in each case). This needs to be investigated and clearly explained in order to understand whether the results can be considered "realistic" or are results of a numerical exercise that mixes two different things.

We hope the updated model calibration can help reviewers find our process-based FFA to be less speculative and more convincing. While ideally model parameters could remain constant regardless of the rainfall dataset used, this is generally not good modeling practice, since rainfall error structures can differ substantially between datasets. For example, due to its much coarser spatial resolution, CPC, even when used in a lumped model, will produce more frequent light rain and lower extremes than Stage IV. Therefore, we believe that calibration for individual input datasets is a necessary evil. Our future research will use distributed physics-based models in place of HBV, and hopefully this is less of an issue in such models.

Major comments 3: For the results in Fig. 5 right panel: Do you use soil moisture years prior to 1990 for the StageIV process-based approach? Also, you should apply the Bull. 17B for the two periods (1933-1989 and 1990-2016) and add them on the graph for comparison.

We did not use the soil moisture prior to 1990 for the Stage IV-based simulation. The antecedent conditions for Stage IV-based simulation are only sampled from continuous simulation of Stage IV period, which is 2002-2016. We have not applied the Bull.17B method to annual daily streamflow maxima for 1933-1989 period because we have not investigated any RainyDay-based simulation for the corresponding time. However, we have added a supplementary plot showing the CPC, Stage IV and Bull.17B based FFA for the modern time (2002-2016), similar to what this reviewer and reviewer 1 suggest.

Supplementary Fig. 1 shows that process-based FFA using CPC precipitation from 2002-2016 closely resembles the Stage IV-based FFA, suggesting that rainfall differences, rather than model structures, are the primary drivers of the differences in this figure. It also shows two features that result using CPC data. First, the extreme tail is underestimated, relative to the Stage IV-based simulations and the statistical approach. CPC is known to contain errors in the extreme tail, due to gage undercatch, insufficient gage density to properly sample convective rain cells, and spatial averaging of such cells over large areas, which effectively reduces peak rainfall depths. Second, CPC overestimates the magnitude of more frequent events. This is likely the result of its coarse spatial resolution, which will "smear" rainfall over larger areas (i.e. entire ~600 km2 grid cells) when it should in reality be more localized. This would serve to increase the likelihood of rainfall over the watershed, albeit at relatively lower depths/intensities. Thus, if one is to restrict the time period of the rainfall data to recent years (for example, the 2002-2016 time period for which Stage IV is available), then Stage IV would likely be a better choice.

[Figure]

**Supplementary Figure 1**. Three peak discharge analyses for Turkey River at Garber, IA: RainyDay with Stage IV (2002-2016) and CPC-(2002-2016) rainfall and USGS frequency analyses (1990-2016) using Bulletin 17B methods. Shaded areas denote the ensemble spread (RainyDay-based results) and the 90% confidence intervals (Bulletin 17B-based analysis), respectively. All observed annual daily streamflow maxima from 1990 to 2016 are shown in black dots.

Minor comments 1: P1, L18: "a watershed that is undergoing significant climatic… change". Is the climatic change at the scale of the watershed only? Consider revising.

We have revised this sentence to:

The methodology is applied to the Turkey River watershed in the Midwestern United States, which is undergoing significant climatic and hydrologic change.

Minor comments 2: P16, L2: "but higher estimates" should be "but gives higher estimates"?

Correct. We have modified that sentence to "but yields higher estimates for rarer events".

Minor comments 3: Fig.6: Improve caption. What is the upper and what the lower panel?

This figure has been updated.

Minor comments 4: P18L13: "processes in her" should be "processes in his/her"

We have updated the text.

---

## Author Comment (AC3) · 14 Feb 2019

*Replies to the comments of Anonymous Referee #3*

Responses are provided in blue and proposed revision are in Red. Original reviewer comments are in black. Line and page numbers refer to the original manuscript.

Based upon comments from all three reviewers, we have revisited our model calibration procedure and have been able to obtain acceptable performance from the snowpack routine. This involved a "2-step" calibration process in which warm season processes are calibrated first, and then "warm season parameters " are held constant during subsequent calibration of snowpack-related parameters. This recalibration of HBV is done using both CPC and Stage IV rainfall. We have also added a section on model validation to the revised manuscript, again based on comments from all three reviewers requesting additional validation results. Since all three reviewers provided critiques on these topics, we discuss these two changes before addressing specific comments from individual reviewers.

We have revised model calibration part in the original manuscript, P9, line 15-24, to:

We calibrated the HBV models using both CPC and Stage IV rainfall, and most parameters are the same for CPC- and Stage IV-based models, except for three snow routine parameters (TT, CFMAX, SFCF) and three recession coefficients (K0, K1, K2), allowing for the variability of model parameters for different climate conditions. For each model setup, we first calibrated the model with snowpack routine "turned off" (by setting TT parameter to a very low value) to obtain parameters that can simulate summer floods adequately. Then, keeping these optimized non-snow routine parameters unchanged, we calibrated the snow routine parameters.

To determine the optimized model parameter sets in each procedures, we followed the Genetic Algorithm and Powell (GAP) optimization method as presented by Seibert (2000), which is briefly summarized here. First, 5000 parameter sets are randomly generated from a uniform distribution of the values of each parameter (Table 1), which were then applied to the HBV model in order to maximize Kling Gupta Efficiency (Gupta et al., 2009) of simulated daily discharge. After the GAP has finished, the optimized parameter set were fine-tuned using Powell's quadratic convergent method (Press, 1996) with 1000 additional runs. Lastly, the optimized parameter set was manually adjusted to improve the fits between observed and simulated annual peak flow (see Lamb, 1999). More elaborate calibration and uncertainty estimation procedures such as Generalized Likelihood Uncertainty Estimation (GLUE; Beven and Binley, 1992; Beven, 1993; Beven and Binley, 2014) could be used, but are outside the scope of our study.

After calibration, HBV (two different parameter sets) was used to perform CS with historical CPC and Stage IV rainfall and temperature data to derive long-term simulated soil moisture and snowpack values, which are usually difficult to obtain via measurement. We "pair" samples of these initial conditions with synthetic rainfall events, as described in Sect. 4.2 and Sect. 4.3.

**Table 1.** Overview of HBV model parameters and prior parameter boundaries.

| Parameter | Description | Units | Min value | Max value |
|---|---|---|---|---|
| Snow Routine | | | | |
| TT | Threshold temperature for liquid and solid precipitation | °C | -3 | 3 |
| CFMAX | Degree-day factor | mm d$^{-1}$°C$^{-1}$ | 0.5 | 4 |
| SFCF | Snowfall correction factor | - | 0.5 | 1.2 |
| CFR | Refreezing coefficient | - | 0.01 | 0.1 |
| CWH | Water holding capacity of the snow storage | - | 0.1 | 0.3 |
| Soil Moisture Routine | | | | |
| FC | Maximum soil moisture storage (field capacity) | mm | 100 | 550 |
| LP | Relative soil water storage below which AET is reduced linearly | - | 0.3 | 1 |
| BETA | Exponential factor for runoff generation | - | 1 | 5 |
| Response Routine | | | | |
| PERC | Maximum percolation from upper to lower groundwater box | mm d$^{-1}$ | 0 | 10 |
| UZL | Threshold of upper groundwater box | mm | 0 | 50 |
| K0 | Recession coefficient 0 | d$^{-1}$ | 0.5 | 0.9 |
| K1 | Recession coefficient 1 | d$^{-1}$ | 0.15 | 0.5 |
| K2 | Recession coefficient 2 | d$^{-1}$ | 0.01 | 0.15 |
| Routing Routine | | | | |
| MAXBAS | Length of triangular weighting function | d | 1 | 2.5 |

We have also added "Section 5.2 Model Validation" by modifying the original paper, P13-14, to:

**5.2 Model Validation**

We validated the performance of HBV continuous simulation with respect to flood seasonality, frequency of annual daily discharge maxima, and normalized peak flow (i.e. the simulated or observed daily discharge divided by the 2-year flood), using both Stage IV and CPC as precipitation inputs (Fig. 4). We also validated two structures: one with and the other without the HBV snowpack module. The purpose for this latter validation effort is to highlight the importance of proper process representation (and subsequent validation) in process-based FFA.

Simulated flood seasonality varies substantially during the CPC period of record (1948-2016) depending on the inclusion of the snowpack routine. Differences are less for the Stage IV period of record (2002-2016), due to the decreasing role of snowpack in deriving the floods in recent years (Fig. 4a). In both cases, the seasonality of flooding simulated using HBV is improved with the inclusion of the snowpack module, with a higher (lower) frequency of springtime (summertime) floods which more closely resembles observations. Empirical (i.e. plotting position-based) distributions for the simulated annual daily discharge maxima are mostly within the 90% confidence interval (obtained by nonparametric bootstrap) of the observations (Fig.

4b). The CPC-based simulations differ considerably depending on the inclusion of the snowpack module for more common events, but differences in simulated maxima vanish as flood magnitude increases (e.g. AEP<0.1). This is because the most extreme flood events occur later in the season and are thus independent of snowpack or snowmelt processes. Differences are generally negligible between Stage IV-based simulations with and without snowpack, since floods in this shorter, more recent period are generally driven by summertime thunderstorms. These findings are consistent with the general understanding of the regional seasonality of flooding in the region, as discussed in Sect. 5.1.

We compared all simulated and observed flood peaks that can be associated with a USGS observed daily streamflow value that is at least three times the mean annual daily discharge (Fig. 4c). When associating simulated and observed flood peaks, we look within a 2-day window to allow for modest errors in simulated flood peak timing. All peaks in Fig. 4c are normalized by the median annual (i.e. 2-year) flood, which, as a rule of thumb, can be considered as the "within bank" threshold. Again, HBV with the snowpack routine outperforms the model without it, especially for the small to modest flood events in CPC-based simulations. The model without snowpack routine underestimate the small to modest flood events in two cases due to the neglect of water flux from potential snowmelt. While modest scatter exists in the Stage IV-based simulated peaks, there is no obvious systematic bias with event magnitude when the snowmelt routine is included.

[Figure]

**Figure 1.** HBV model validation for flood seasonality **(a)**, frequency of annual max. daily discharge **(b)** and normalized peak flow (c). For each panel, the corresponding model validation is performed against CPC- (1948-2016) and StageIV-based (2002-2016) simulation and the results derived from HBV model with (without) snowpack routine are shown in blue (red). The 90% confidence interval for observed max. daily discharge (empirical distribution) is derived using the bootstrapping approach. Peak discharge is defined as a data point with USGS observed value that is at least three times the average observations, and peak discharge are normalized by the median of annual daily discharge maxima (i.e. the 2-year flood). Straight black lines indicate 1:1 correspondence, while dashed lines indicate the envelope within which the modeled values are within 50% of observed.

We also validate HBV's snowpack routine using observed GHCN daily snow depth for two simulation periods (Fig. 5a, 5b) and using USGS daily streamflow observations for Stage IV-based period (Fig. 5c). Because of their differing spatial resolutions and physical representations, point-scale GHCN daily snow

depths cannot be directly or quantitatively compared to the watershed-scale snow water equivalent simulated by HBV. Therefore, we validate the snowpack simulation in terms of the snowpack occurrence, defined as the number of occurrences where snow is present on a particular date divided by the total number of years in the historical record. For example, there are 50 days where snowpack is present on January 1st in the 69-year period from 1948-2016, based on GHCN observations and thus the corresponding occurrence rate is 0.72 (50 divided by 69). The HBV model with the snowpack routine captures the central tendency of observed snowpack dynamics, showing that snowpack frequently exists from early November to mid-February, with frequency of snow decreasing from late February until disappearing in early April.

[Figure]

**Figure 2.** The comparison of percent of days with snowpack present between observations and simulations (a, b) and hydrograph validation for StageIV-based simulation (c). For each day within a year, the percent of snowpack existing days is calculated as the ratio of the number of years when snowpack is present to the total years (69 years for CPC and 15 years for StageIV). Observed and simulated hydrograph are normalized by the median annual flood, which is indicated by the dashed blue line.

Model hydrograph validation is provided in Fig. 5c for the Stage IV period (2002-2016), when major flooding occurred throughout Iowa. Model performance shows no obvious evidence of systematic bias in the streamflow simulations. Although flood seasonality derived by Stage IV-based simulation differs slightly from observations (Fig. 4b), these mismatches are associated with flood events smaller than the median annual flood (blue dash line in Fig. 5c). Stage IV-based simulations do not show bias flood magnitude in late summer. In other words, remaining biases in terms of flood seasonality generally

This combination of continuous and event based modelling is a quite novel idea and provides a flexible framework for DFFA. The application of the methods seems sound, the research is done systematically and the paper reads quite well. However, I do have some concerns regarding the selection of the hydrological model, the selection of two precipitation data sets and some of the conclusions. I will detail these below in the major comments, followed by some minor comments. The paper is worth to be published after major revision.

We thank the reviewer for these useful critiques, which have been very helpful in improving the paper. We address these issues more deeply in specific responses, but generally speaking: 1.) in the revised manuscript, we have reintroduced the snowpack routine in the HBV and calibrate and validate the model more carefully. We discussed the model validation with respect to the flood seasonality, peak flow, snowpack, and hydrographs. 2.) we discuss the limitations of CPC precipitation data and the reason why we include the Stage IV precipitation data in this process-based FFA framework. 3.) we provide a short summary of the pros and cons of the proposed FFA framework.

Major comments 1: The selection of the lumped HBV model is not plausible to me, especially given that a) the snow routine is not working and b) the high resolution StageIV rainfall data cannot be utilized by this lumped model.

Since we have updated the HBV model by including the snowpack routine and validated the model as shown in the beginning of this response, we hope the reviewer finds the selection of the lumped HBV model to be more convincing. It also should be noted that, the process-based FFA methodology employed in this study could be coupled with other (sophisticated) hydrologic models, as we mentioned in the original manuscript, P9, line 10, and, in fact, that is our next research direction. Nonetheless, after decades of research, lumped models have still proven to be very useful in a variety of hydrologic fields including flood applications and research. One challenge that we faced in this study was how to quickly implement and evaluate modifications and additions to the methodology, which can be much slower and more challenging using a more sophisticated distributed model.

We respectfully disagree that the Stage IV rainfall data cannot be utilized by a lumped model. Regardless of model choice, Stage IV precipitation data is generally better than CPC data in the study region, in terms of accuracy-this is evident, for example, in the fact that the satellite precipitation community routinely uses Stage IV and related gage-corrected radar products, rather than CPC, to validate satellite rainfall estimates. CPC is known to contain errors in the extreme tail, due to gage undercatch, insufficient gage density to properly sample convective rain cells, and spatial averaging of such cells over large areas, which effectively reduces peak rainfall depths. Second, CPC overestimates the magnitude of more frequent events. This is likely the result of its coarse spatial resolution, which will "smear" rainfall over larger areas (i.e. entire ~600 km2) grid cells when it should be more localized. This would serve to increase the likelihood of rainfall over the watershed, albeit at relatively lower depths/intensities. Thus, if one is to restrict the time period of the rainfall data to recent years (for example, the 2002-2016 time period for which Stage IV is

available), then Stage IV would likely be better. It is true that the lumped model cannot "leverage" the rainfall spatial structure embedded in Stage IV, but it still benefit from its improved accuracy.

Major comments 2: The application of two rainfall data sets is not plausible and also quite confusing for the reader since a) the Stage IV rainfall data observation period (2002-2016) is covered also by the CPC rainfall data observation period (1948-2016), b) a lumped hydrological model cannot really benefit from high resolution rainfall data (see 1) and c) the hydrological simulation results for both rainfall data sets seem to be very similar (as the authors state on page 16, lines 12-13). I would recommend to do all the simulations with the CPC rainfall if the hydrological model is not changed. If a more suitable hydrological model is selected the two data sets might be kept in the study but the differences in hydrological response using the two data sets for the same time period (2002-2016) need also to be demonstrated and discussed.

We feel that including the Stage IV-based simulation in this case study is important in two respects: 1.) As mentioned in the response to comment 1, we believe the Stage IV precipitation data has high accuracy than CPC. As an aside, this belief that Stage IV is preferable to other datasets when long records are not required is widely shared in the satellite precipitation validation community, where Stage IV is often used as a validation dataset. 2.) We also want to highlight that using only 15 years of rainfall records, our process-based approach can produce accurate estimates of "present-day" flood frequency.

In addition, we have analyzed two CPC-based results from 1948-2016 and 2002-2016 to demonstrate how the changes in flood agents affect the FFAs. We have added the following part to Sect.5.3, P17, line 21 of the original manuscript.

To demonstrate that the discrepancies between the process-based FFA results generated using CPC and using StageIV are driven by changes in flood agents, rather than by differences in model structure (i.e. parameter values), we compared FFA results generated using CPC-based for 1948-2016 and 2002-2016, in terms of event rainfall, initial soil moisture, flood type and peak magnitude (Fig. 8). From 2002-2016 (Fig. 8b), there are fewer flood events driven by snowmelt or rain-on-snow but more driven by rainfall, particularly large magnitude flood events (over 1000 m3/s). In addition, some of the rainfall driven floods (upper left of Fig. 8b) from 2002-2016 indicates high initial soil moisture, which are in accordance with the significant increasing trend of annual precipitation (Table 2). In general, changes in individual flood agents and their interactions can affect flood frequency. Process-based approaches can help illuminate these changes.

[Figure]

Figure 8. The simulated flood magnitude using CPC rainfall during 1948-2016 (a) and 2002-2016 (b) period, and corresponding antecedent conditions sampled from the continuous simulation. The blue triangles represent the snow related flood events (e.g. snowmelt or rain on snow) and grey dots represents the non-snow related flood events (e.g. rainfall driven). The size of the triangles or dots indicate the antecedent soil moisture with higher value in larger shape. The black dash line indicates the 1000m3/s flood magnitudes.

Major comments 3: The application of a model without snow routine for a catchment with significant snow processes doesn't make sense to me. This way the advantage of process based flood frequency analysis (FFA) is partly lost; obtaining the correct hydrological response for the wrong reason is not satisfying. I am not convinced that the non-stationarity in seasonality is only due to changed soil moisture conditions from rainfall. Temporarily shifted snow dynamics might play a role as well

After taking the reviewers' comments into account very seriously, we recalibrate our model with snowpack routine "turned on" and validate it with respect to flood seasonality, hydrograph, normalized peak flow and snowpack. We finally conclude that the snowpack routine of HBV is indeed important in this study region.

Major comments 4: I would be careful with the conclusion, that only with this DFFA method nonstationarity in seasonality can be handled well. Also, non-stationary seasonal FFA approaches are available employing mixed distributions for getting final design values. This needs to be briefly discussed.

We appreciate the comment. Certainly seasonality could be considered using other approaches, though mixture distribution approaches may still not elucidate the fundamental drivers that "shape" flood frequency, even if they can provide good end results. We are not aware of such approaches being used in widespread practice, at least in the United States. Nonetheless, we had added a brief comment in this regard to the conclusions in acknowledgement of this criticism.

We have revised the first paragraph of Section 6 on P21, line 13-15, to :

It must be noticed that the statistical approaches coupling with flood seasonality indices can also investigate the impacts of seasonality on FFA and improve the flood frequency estimation in a regional scale (Ouarda et al., 2006). Our aim is to estimate flood quantiles by reconstructing meteorological and hydrological processes and their interactions, providing an alternative approach which is also well-suited to nonstationary environments (see also Sivapalan and Samuel, 2009).

Major comments 5: This combination of continuous and event based modelling is a good idea. However, there is an important limitation which should at least be mentioned. The framework provides only one possible realization of initial conditions. Nature is more variable. Stochastic rainfall models producing continuous rainfall don't pose this limitation on hydrology.

Each event-based simulation is randomly paired with initial conditions drawn from a continuous simulation (15 years in the case of Stage IV, 69 years for CPC). Thus, we would argue that a large number of possible realizations of initial conditions are used. We would direct the reviewer to Section 4.3. If the reviewer finds this description incomplete, we would appreciate suggestions for how we can make this point more clear. Though we have not tested rigorously, we would guess that relatively short records (say, 15 years) of continuous simulations are sufficient to obtain enough variability in initial conditions. Compared with rainfall, soil moisture (which is bounded between 0 and saturation) and springtime snowpack have thinner tails and thus easier to represent in our framework by sampling from relatively short continuous simulation.

We agree that continuous stochastic rainfall models also have the ability to produce a wide range of pre-event conditions, though it is likely nontrivial to properly calibrate their seasonality with respect to the extreme tail of precipitation-demanding long training datasets.

Minor comments 1: Page 2, line 4: This sentence is confusing. I am assuming you mean '... statistical analysis of observed streamflow, design storms !and! continuous simulation !or! other so called "derived" or "process based" methods'.

Correct. We have modified this sentence to:

Most existing FFA methods belong to one of three approaches: statistical analysis of streamflow observations, design storms, and continuous simulation or other so-called "derived" or "process-based" methods.

Minor comments 2: Page 4, lines 15-17: This sentence seems not to be complete.

We apologize for this. We have revised this sentence to:

Wright et al. (2014a) discusses additional design storm shortcomings including time of concentration concepts, in greater detail, while also pointing out that design storm approaches (like other hydrological model-based FFA) can incorporate future projections in land use and rainfall more explicitly than can statistical discharge-based methods.

Minor comments 3: Page 10, steps 3 and 4: I would stress that the 30 storms per year are randomly transposed over the domain, only sometimes hitting the catchment and sometimes not. They are not all transposed on the catchment, which would lead to an overestimation of the flood frequency. The reader not familiar with your method might misunderstand that.

The reviewer is correct. We have added this sentence to P10, line 22.

It must be noted that some of the *k* transposed storms may not "hit" Turkey River watershed, and thus their calculated watershed rainfall are zero.

Minor comments 4: Page 11, lines 8-9: The selection of the largest event per year for FFA might also be misunderstood. Here, it also needs to be considered that many of the 30 events do not produce any flood if they do not hit the catchment (see comment 3).

We hope the response to previous comments also addresses this one.

Minor comments 5: Page 14: line 2: Should it not be "… but overestimates for pe<0.3 …"

We assume the reviewer mean Page 16, line 2. We have revised this sentence to:

The Stage IV-based flood frequency curve agrees reasonably well with the discharge-based FFA for $p_e > 0.3$ (left panel of Fig. 6), but yields higher estimates for rarer events.

Minor comments 6: Fig. 5: Why did you select the period 1990 – 2016 and not 1980 or 1970 as starting year? This needs to be justified.

We have not performed any statistical test (e.g. Pettitt test) to determine this change point. However, an "eyeball test" of annual daily discharge maxima (Fig. 1a) from the original manuscript indicates the apparent elevated flood activity during 1990-2016 period. Our arguments do not hinge on a precise determination of when floods in Turkey River began to change, which in any event has likely been a gradual change.

Minor comments 7: Fig. 5: I would also add a statistical analysis (Bull 17.b) for the contemporary period (1990-2016) for comparison.

We have added a supplementary plot showing the CPC, Stage IV and Bull.17B based FFA for the modern time (2002-2016), as other reviewers have suggested.

Supplementary Fig. 1 shows that process-based FFA using CPC precipitation from 2002-2016 closely resembles the Stage IV-based FFA, suggesting that rainfall differences, rather than model structures, are the primary drivers of the differences in this figure. It also shows two features that result using CPC data. First, the extreme tail is underestimated, relative to the Stage IV-based simulations and the statistical approach. CPC is known to contain errors in the extreme tail, due to gage undercatch, insufficient gage density to properly sample convective rain cells, and spatial

averaging of such cells over large areas, which effectively reduces peak rainfall depths. Second, CPC overestimates the magnitude of more frequent events. This is likely the result of its coarse spatial resolution, which will "smear" rainfall over larger areas (i.e. entire ~600 km2 grid cells) when it should be more localized. This would serve to increase the likelihood of rainfall over the watershed, albeit at relatively lower depths/intensities. Thus, if one is to restrict the time period of the rainfall data to recent years (for example, the 2002-2016 time period for which Stage IV is available), then Stage IV would likely be better.

[Figure]

**Supplementary Figure 1**. Three peak discharge analyses for Turkey River at Garber, IA: RainyDay with Stage IV (2002-2016) and CPC-(2002-2016) rainfall and USGS frequency analyses (1990-2016) using Bulletin 17B methods. Shaded areas denote the ensemble spread (RainyDay-based results) and the 90% confidence intervals (Bulletin 17B-based analysis), respectively. All observed annual daily streamflow maxima from 1990 to 2016 are shown in black dots.

Minor comments 8: Fig. 6: There is no description neither in legend nor in figure caption about the source of the two figures. I assume they stem from different precipitation data sets.

We have updated this figure.

---

## Author Response (AR1)

Manuscript No: hess-2018-513 Title: Process-Based Flood Frequency Analysis in an Agricultural Watershed Exhibiting Nonstationary Flood Seasonality Authors: Guo Yu et al.

Dear Editor and Reviewers,

We would like to thank the Editor and three anonymous reviewers for providing critical and helpful comments. The central goal has been to create a well-organized paper that highlights the opportunity and challenge with process-based flood frequency analysis (FFA) approaches. The reviewers have contributed greatly to that goal and we have carefully considered each of their criticisms.

The changes that we have made on the basis of these criticisms constitute a major overhaul of the original manuscript, including:

- 1) revisit our model calibration procedure and include the snowpack routine to obtain acceptable performance;
- 2) add a section on model validation to the revised manuscript;
- 3) highlight the impacts of changing flood seasonality on FFA by adding Fig.8 to the revised manuscript;
- add the analysis of pros and cons on process-based FFA approaches to the conclusion section;
- 5) revised/restructured the focus of the introduction and conclusion to clarify the objectives of this study.

Therefore, we would appreciate that the reviewers grant us another through reading.

Sincerely,

Guo Yu

**Common Responses to three reviewers**

Responses are provided in blue and proposed revision are in Red. Original reviewer comments are in black. Line and page numbers refer to the original manuscript.

Based upon comments from all three reviewers, we have revisited our model calibration procedure and have been able to obtain acceptable performance from the snowpack routine. This involved a "2-step" calibration process in which warm season processes are calibrated first, and then "warm season parameters " are held constant during subsequent calibration of snowpack-related parameters. This recalibration of HBV is done using both CPC and Stage IV rainfall. We have also added a section on model validation to the revised manuscript, again based on comments from all three reviewers requesting additional validation results. Since all three reviewers provided critiques on these topics, we discuss these two changes before addressing specific comments from individual reviewers.

We have revised model calibration part in the original manuscript, P9, line 15-24, to:

We calibrated the HBV models using both CPC and Stage IV rainfall, and most parameters are the same for CPC- and Stage IV-based models, except for three snow routine parameters (TT, CFMAX, SFCF) and three recession coefficients (K0, K1, K2), allowing for the variability of model parameters for different climate conditions. For each model setup, we first calibrated the model with snowpack routine "turned off" (by setting TT parameter to a very low value) to obtain parameters that can simulate summer floods adequately. Then, keeping these optimized non-snow routine parameters unchanged, we calibrated the snow routine parameters.

To determine the optimized model parameter sets in each procedures, we followed the Genetic Algorithm and Powell (GAP) optimization method as presented by Seibert (2000), which is briefly summarized here. First, 5000 parameter sets are randomly generated from a uniform distribution of the values of each parameter (Table 1), which were then applied to the HBV model in order to maximize Kling Gupta Efficiency (Gupta et al., 2009) of simulated daily discharge. After the GAP has finished, the optimized parameter set were fine-tuned using Powell's quadratic convergent method (Press, 1996) with 1000 additional runs. Lastly, the optimized parameter set was manually adjusted to improve the fits between observed and simulated annual peak flow (see Lamb, 1999). More elaborate calibration and uncertainty estimation procedures such as Generalized Likelihood Uncertainty Estimation (GLUE; Beven and Binley, 1992; Beven, 1993; Beven and Binley, 2014) could be used, but are outside the scope of our study.

After calibration, HBV (two different parameter sets) was used to perform CS with historical CPC and Stage IV rainfall and temperature data to derive long-term simulated soil moisture and snowpack values, which are usually difficult to obtain via measurement. We "pair" samples of these initial conditions with synthetic rainfall events, as described in Sect. 4.2 and Sect. 4.3.

| Parameter             | Description                                                     | Units                        | Min value | Max value |
|-----------------------|-----------------------------------------------------------------|------------------------------|-----------|-----------|
| Snow Routine          |                                                                 |                              |           |           |
| TT                    | Threshold temperature for liquid and solid precipitation        | °C                           | -3        | 3         |
| CFMAX                 | Degree-day factor                                               | mm $d^{-1}$ °C -1 | 0.5       | 4         |
| SFCF                  | Snowfall correction factor                                      | -                            | 0.5       | 1.2       |
| CFR                   | Refreezing coefficient                                          | -                            | 0.01      | 0.1       |
| CWH                   | Water holding capacity of the snow storage                      | -                            | 0.1       | 0.3       |
| Soil Moisture Routine |                                                                 |                              |           |           |
| FC                    | Maximum soil moisture storage (field capacity)                  | mm                           | 100       | 550       |
| LP                    | Relative soil water storage below which AET is reduced linearly | -                            | 0.3       | 1         |
| BETA                  | Exponential factor for runoff generation                        | -                            | 1         | 5         |
| Response Routine      |                                                                 |                              |           |           |
| PERC                  | Maximum percolation from upper to lower groundwater box         | $mm d^{-1}$                  | 0         | 10        |
| UZL                   | Threshold of upper groundwater box                              | mm                           | 0         | 50        |
| K0                    | Recession coefficient 0                                         | d -1              | 0.5       | 0.9       |
| K1                    | Recession coefficient 1                                         | d -1              | 0.15      | 0.5       |
| K2                    | Recession coefficient 2                                         | d -1              | 0.01      | 0.15      |
| Routing Routine       |                                                                 |                              |           |           |
| MAXBAS                | Length of triangular weighting function                         | d                            | 1         | 2.5       |

Table 1. Overview of HBV model parameters and prior parameter boundaries.

We have also added "Section 5.2 Model Validation" by modifying the original paper, P13-14, to:

**5.2 Model Validation**

We validated the performance of HBV continuous simulation with respect to flood seasonality, frequency of annual daily discharge maxima, and normalized peak flow (i.e. the simulated or observed daily discharge divided by the 2-year flood), using both Stage IV and CPC as precipitation inputs (Fig. 4). We also validated two structures: one with and the other without the HBV snowpack module. The purpose for this latter validation effort is to highlight the importance of proper process representation (and subsequent validation) in process-based FFA.

Simulated flood seasonality varies substantially during the CPC period of record (1948-2016) depending on the inclusion of the snowpack routine. Differences are less for the Stage IV period of record (2002-2016), due to the decreasing role of snowpack in deriving the floods in recent years (Fig. 4a). In both cases, the seasonality of flooding simulated using HBV is improved with the inclusion of the snowpack module, with a higher (lower) frequency of springtime (summertime) floods which more closely resembles observations. Empirical (i.e. plotting position-based) distributions for the simulated annual daily discharge maxima are mostly within the 90% confidence interval (obtained by nonparametric bootstrap) of the observations (Fig.

4b). The CPC-based simulations differ considerably depending on the inclusion of the snowpack module for more common events, but differences in simulated maxima vanish as flood magnitude increases (e.g. AEP

---

## Author Response (AR2)

Manuscript No: hess-2018-513
Title: Process-Based Flood Frequency Analysis in an Agricultural Watershed Exhibiting Nonstationary Flood Seasonality
Authors: Guo Yu et al.

Dear Editor and Reviewers,

We would like to thank you again for all of your constructive comments which have greatly improved the quality of this paper. We have modified all the technical issues as suggested by the editor. For Figure 1.a, we have double checked the map and it should be correct. The seemingly elongated map of United States is caused by Plate Carre projection since any projections distort the original map to some degree.

Besides these technical issues, we also addressed one non-technical issue by adding following sentences to page 12, line 6, as suggested by the editor.

"*As mentioned in Step 5 of the SST procedure (Section 4.2), each synthetic rainfall event is randomly paired with seasonally-appropriate initial conditions (soil moisture, snowpack) and air temperature drawn from the continuous simulation (15 years in the case of Stage IV; 69 years for CPC). This creates combinations of initial conditions and forcing that in principle reflect the true variability of these processes.*"

Sincerely,

Guo Yu